# Cryo-EM structures of the caspase-activated protein XKR9 involved in apoptotic lipid scrambling

Monique S Straub, Carolina Alvadia, Marta Sawicka, Raimund Dutzler*

Department of Biochemistry, University of Zurich, Zurich, Switzerland

**Abstract** The exposure of the negatively charged lipid phosphatidylserine on the cell surface, catalyzed by lipid scramblases, is an important signal for the clearance of apoptotic cells by macrophages. The protein XKR9 is a member of a conserved family that has been associated with apoptotic lipid scrambling. Here, we describe structures of full-length and caspase-treated XKR9 from *Rattus norvegicus* in complex with a synthetic nanobody determined by cryo-electron microscopy. The 43 kDa monomeric membrane protein can be divided into two structurally related repeats, each containing four membrane-spanning segments and a helix that is partly inserted into the lipid bilayer. In the full-length protein, the C-terminus interacts with a hydrophobic pocket located at the intracellular side acting as an inhibitor of protein function. Cleavage by caspase-3 at a specific site releases 16 residues of the C-terminus, thus making the pocket accessible to the cytoplasm. Collectively, the work has revealed the unknown architecture of the XKR family and has provided initial insight into its activation by caspases.

***For correspondence:**
dutzler@bioc.uzh.ch

**Competing interests:** The authors declare that no competing interests exist.

## Introduction

The lipid distribution in the two leaflets of the plasma membrane of a eukaryotic cell is asymmetric with phosphatidylethanolamine (PE) and the anionic lipids phosphatidylserine (PS) and phosphoinositol (PI) located in the inner leaflet and phosphatidylcholine (PC) and sphingomyelin (SM), in the outer leaflet of the bilayer (*Balasubramanian and Schroit, 2003*; *Bevers and Williamson, 2016*; *Kobayashi and Menon, 2018*). This asymmetry is established and maintained by ATP-dependent active transport processes (*Leventis and Grinstein, 2010*; *Sebastian et al., 2012*). Although preserved under resting conditions, the controlled breakdown of the lipid asymmetry changes the properties of the membrane and is exploited in different processes ranging from signal transduction to membrane fusion (*Nagata et al., 2016*; *Whitlock and Hartzell, 2017*). Asymmetry breakdown is catalyzed by lipid channels termed scramblases, which catalyze the bidirectional diffusion of lipids with poor specificity by lowering the barrier for lipid flip-flop between the two leaflets of the membrane (*Brunner et al., 2016*; *Nagata et al., 2020*; *Pomorski and Menon, 2006*). The resulting equilibration causes the exposure of negatively charged PS to the outside, which serves as a receptor for proteins that initiate specific signaling cascades. In case of apoptosis, PS exposure serves as an eat-me signal for macrophages, leading to the engulfment and clearance of apoptotic cells (*Segawa and Nagata, 2015*). To prevent endogenous activity, which would be deleterious to cells, scramblases residing at the plasma membrane are tightly regulated. While incapable of catalyzing lipid transport under resting conditions, they become activated by specific stimuli, which lead to conformational rearrangements of the protein (*Nagata et al., 2020*). Active scramblases are believed to function by a 'credit card-like mechanism', by offering suitable pathways for lipid headgroups to cross the hydrophobic membrane core, whereas their apolar tails reside in the membrane (*Pomorski and Menon, 2006*).

Lipid scrambling is catalyzed by different proteins with the currently best characterized examples belonging to the TMEM16 and XKR families (*Suzuki et al., 2013*; *Suzuki et al., 2010*). TMEM16 scramblases are activated in response to the increase of the intracellular $Ca^{2+}$ concentration. Their mechanism of activation and lipid transport are best understood for the two fungal homologs nhTMEM16 and afTMEM16 (*Brunner et al., 2014*; *Falzone et al., 2019*; *Kalienkova et al., 2019*; *Kalienkova et al., 2021*). These proteins contain polar membrane-spanning furrows of appropriate size, which are blocked in the absence of $Ca^{2+}$ but become exposed to the membrane upon $Ca^{2+}$ binding to offer suitable pathways for polar headgroups across the membrane, as predicted by the 'credit card mechanism'. The fungal TMEM16 proteins were also found to disturb the organization of lipids in their vicinity, likely by destabilizing the bilayer and thus further decreasing the barrier for lipid flip-flop (*Falzone et al., 2019*; *Kalienkova et al., 2019*).

Although scramblases of the TMEM16 family have been associated with diverse processes such as blood coagulation and cell-fusion, they are not the long sought-after proteins that are responsible for the exposure of PS during apoptosis. The molecular identity of an apoptotic scramblase was first identified in 2013 when the human protein XKR8 and its *Caenorhabditis elegans* homolog Ced-8 were assigned as the scramblases that are responsible for PS exposure in apoptotic cells (*Chen et al., 2013*; *Suzuki et al., 2013*). Both are part of the evolutionarily conserved XKR family of proteins related to the founding member XK1, a molecule with unknown function, which was identified to be associated with the blood-group-specific factor Kell (*Calenda et al., 2006*; *Stanfield and Horvitz, 2000*). In humans, the family contains nine members, of which three, namely the proteins XKR4, 8, and 9, were found to activate scrambling under apoptotic conditions (*Suzuki et al., 2014*). The primary activating step is the cleavage of a C-terminal peptide by the apoptotic peptidase caspase-3 at a specific site that is found in all three proteins (*Suzuki et al., 2013*; *Suzuki et al., 2014*). Besides the irreversible action of caspases, XKR8 was also shown to be activated by phosphorylation (*Sakuragi et al., 2019*).

Another study has revealed the association of XKR8 with accessory proteins such as basigin or its paralog neuroplastin (*Suzuki et al., 2016*). Both type I membraneproteins, containing extracellular IG-domains, facilitate the targeting of XKR8 to the plasma membrane. Based on the altered migration behavior on blue-native gel electrophoresis, the same study also claimed a dimerization of XKR8-basigin complexes upon caspase-3 activation as an essential step underlying activation (*Suzuki et al., 2016*). A similar dimerization was recently postulated for XKR4 (*Maruoka et al., 2021*).

Whereas the central role of certain XKR proteins in apoptotic scrambling and their activation by caspase-3 has been conclusively demonstrated, their molecular architecture and the question whether or how the proteins on their own would catalyze lipid transport has remained elusive. To address these questions, we have here studied the protein XKR9 from *Rattus norvegicus* and determined its structure in its full-length and caspase-cleaved form by cryo-electron microscopy (cryo-EM). Our results reveal the novel architecture of a eukaryotic membrane protein family involved in apoptotic scrambling and provide insight into their activation by caspases.

## Results

### Biochemical characterization of XKR9

In our study, we have selected the XKR9 ortholog from *Rattus norvegicus* (rXKR9) due to its promising biochemical properties. The protein is 373 amino acids long and shares 76% of identical residues with its human ortholog. Following its expression in suspension culture of HEK293S GnTI⁻ cells, purification in the detergent lauryl-maltose-neopentyl-glycol (LMNG), and removal of the N-terminal tag used for affinity purification, we found rXKR9 to elute as monodisperse peak on gel-filtration with an elution volume corresponding to a monomer (*Figure 1A*), which was subsequently quantified by multi-angle light scattering (*Figure 1—figure supplement 1A*). The protein contains a predicted caspase cleavage site, which would remove 16 amino acids from the C-terminus resulting in the activation of scrambling in cells. We have investigated protein digestion after incubation with caspase-3 in detergent solution and found complete cleavage of the C-terminus resulting in the predicted reduction of the molecular weight as confirmed by mass spectrometry. As full-length rXKR9, the truncated protein elutes as a monomer and we did not find strong evidence of dimerization in

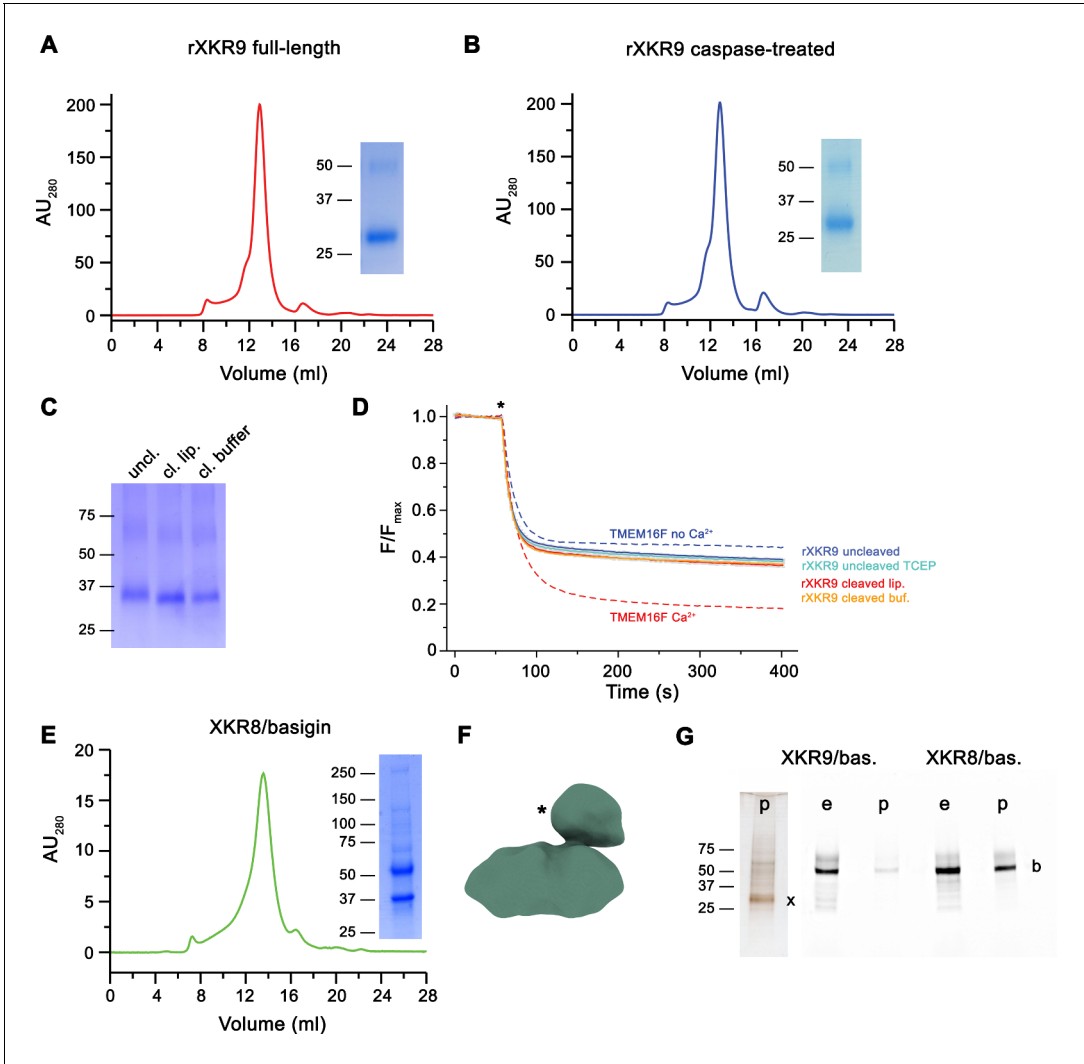

**Figure 1.** Biochemical and functional properties of rXKR9. Size-exclusion chromatogram of full-length (**A**) and caspase-treated (**B**) rXKR9. Insets show SDS–PAGE gel of the pooled and concentrated peak sample. (**C**) SDS–PAGE gel of rXKR9 extracted from different proteoliposome preparations used for lipid scrambling assays. The samples treated with caspase-3 in liposomes (cl. lip.) or detergent solution prior to reconstitution (cl. buffer) show shift toward smaller molecular weight compared to the full-length protein (uncl.). (**D**) Assay to monitor the protein-catalyzed movement of fluorescent lipids between both leaflets of a bilayer. The addition of the reducing agent dithionite to the solution (*) is indicated. Traces of TMEM16F in the absence and presence of 100 µM $Ca^{2+}$ displayed in *Figure 1—figure supplement 1C* are shown as controls for an inactive and active scramblase, respectively (dashed lines). No pronounced activity is observed in any of the liposome preparations containing either full-length or caspase-3-treated samples. Data show mean and standard deviations of three technical replicates. (**E**) Purification of a human XKR8-basigin complex. Left, size-exclusion chromatogram of hXKR8 in complex with basigin containing a GFP fusion. The complex was purified via a tag attached to hXKR8. SDS–PAGE gel showing the Coomassie stained bands of the purified complex (right). (**F**) Low-resolution reconstruction of the hXKR8-basigin complex. The density protruding from the membrane-inserted part (*) corresponds to the extracellular part of basigin. (**G**) In-gel fluorescence showing GFP-tagged basigin (b) in whole-cell extracts (e) and after affinity purification (p) of hXKR8 and rXKR9. Whereas the XKR8 samples show a pronounced basigin band, the XKR9 samples only contain traces of the protein. The presence of XKR9 in the sample (marked with 'x') is shown on silver-stained SDS–PAGE gel (left). The concentration of XKR8 after pulldown is several times lower as judged by its barely detectable band on the same gel (not shown). (**A–C, E, G**) Molecular weight (kDa) of marker proteins on SDS–PAGE gels are indicated.

The online version of this article includes the following source data and figure supplement(s) for figure 1:

**Source data 1.** SDS–PAGE gel of full-length rXKR9 shown in *Figure 1A*.

**Source data 2.** SDS–PAGE gel of caspase-3 treated rXKR9 shown in *Figure 1B*.

**Source data 3.** SDS–PAGE gel of full-length or caspase-3-treated rXKR9 extracted from respective proteoliposomes shown in *Figure 1C*.

**Source data 4.** Normalized individual traces of scrambling experiments displayed in *Figure 1D*.

**Source data 5.** SDS–PAGE gel of the XKR8/basigin complex shown in *Figure 1E*.

**Source data 6.** Silver-stained SDS–PAGE gel of the XKR9 co-expressed with basigin ahown in *Figure 1G*.

*Figure 1 continued on next page*

*Figure 1 continued*

**Source data 7.** In-gel fluorescence of Venus-tagged proteins detected in SDS–PAGE gel of the XKR8 or XKR9 expressed with basigin shown in *Figure 1G*.

**Figure supplement 1.** Biochemical characterization of rXKR9.

**Figure supplement 1—source data 1.** Normalized individual traces of scrambling experiments displayed in *Figure 1—figure supplement 1C*.

**Figure supplement 2.** Preparation of rXKR9-Sb1$^{XKR9}$ complexes.

**Figure supplement 2—source data 1.** SDS–PAGE gel of purified full-length rXKR9 in complex with Sb1$^{XKR9}$ shown in *Figure 1B*.

**Figure supplement 2—source data 2.** SDS–PAGE gel of purified caspase-3 treated rXKR9 in complex with Sb1$^{XKR9}$ shown in *Figure 1B*.

**Figure supplement 2—source data 3.** Mass spectrum of caspase-3-treated XKR9 shown in *Figure 1—figure supplement 2D*.

**Figure supplement 3.** Low-resolution cryo-EM reconstruction of a hXKR8-basigin complex.

detergent solution (*Figure 1B*), which was independently suggested for an XKR8-basigin complex (*Suzuki et al., 2016*) and for XKR4 upon caspase activation (*Maruoka et al., 2021*). To probe whether purified rXKR9 on its own would facilitate lipid scrambling, we have reconstituted the protein into liposomes. Incorporation of the protein was assayed by its re-extraction from harvested proteoliposomes using mild detergents and analysis by size-exclusion chromatography (*Figure 1—figure supplement 1B*). We subsequently investigated the flipping of lipids between the two leaflets of the bilayer by using an assay that was previously established for the characterization of opsin and scramblases of the TMEM16 family (*Brunner and Schenck, 2019*; *Falzone and Accardi, 2020*; *Ploier and Menon, 2016*). The applied assay investigates the quenching of a fluorescent NBD group attached to one of the tails of a lipid that was added to the liposomes as minor component. The reduction of fluorescence proceeded by addition of the membrane-impermeable reagent dithionite to the outside, which bleaches NBD-containing lipids located on the outer leaflet of the bilayer. In our assays, we investigated the scrambling activity in proteoliposomes containing either full-length rXKR9, a caspase-3-treated form of the protein where the C-terminus was cleaved-off before reconstitution, or proteoliposomes of full-length rXKR9 where the accessible C-terminus of proteins in inside-out orientation was removed upon addition of caspase-3 to the medium. In latter case, reducing conditions were required for the protease to be active, which was achieved by supplementation of the buffer with 1 mM of TCEP, which did neither reduce the labeled lipids nor interfere with liposome integrity. Whereas in both cases, cleavage was detected on SDS–PAGE (*Figure 1C*), the assay used to investigate the integrity of the reconstituted protein indicated that removal of the C-terminus might have compromised the stability of rXKR9, which is particularly pronounced for protein that was incubated with caspase-3 prior to reconstitution. After re-extraction of the digested protein from proteoliposomes, we found for cleaved protein a decreasing peak height at elution volumes corresponding to monomers and pronounced peaks eluting at the void volume of the column (*Figure 1—figure supplement 1B*). In our liposome-based assays, we did not find strong evidence of scrambling activity irrespective of the cleavage of the C-terminus (*Figure 1D*, *Figure 1—figure supplement 1C*), thus suggesting that the protein does not promote lipid scrambling under the applied conditions, which might be the consequence of the lack of an essential factor required for the protein to be active.

We also were interested in the interaction of XKR9 with accessory proteins, since the human homolog hXKR8 was shown to form a complex with the type I membrane protein basigin (*Suzuki et al., 2016*). We thus engaged in biochemical studies investigating the co-expression of either protein with their corresponding species-specific basigin orthologs. Whereas in case of hXKR8, we find robust complex formation with basigin, which is also manifested in a low-resolution cryo-EM reconstruction (*Figure 1E,F*) and in a recently determined structure of the complex at high resolution (*Sakuragi et al., 2021*), we do not find evidence for the formation of a similar complex with rXKR9 (*Figure 1G*), thus refuting the assumption that basigin might be an essential interaction partner of the protein.

## Structure determination of XKR9-sybody complexes

Although rXKR9 did not show any scrambling activity in vitro, its high similarity to hXKR8 (with both proteins sharing 29% of identical and 53% of homologous residues) and its susceptibility to caspase-3 makes it an attractive target to elucidate the unknown architecture of the XKR family and the molecular consequences of its proteolytic activation. As a monomeric protein containing 373

residues only, rXKR9 is on the small side for structural studies by cryo-EM and we thus attempted to increase its size by forming complexes with suitable binding proteins. To this end, we engaged in the selection of nanobodies from synthetic libraries (termed sybodies) targeting purified rXKR9 by a combination of ribosome- and phage-display (*Zimmermann et al., 2018*; *Zimmermann et al., 2020*). Our attempts allowed us to identify several sybodies. From the selected set of binders, we chose the sybody Sb1^XKR9 to generate complexes with full-length and cleaved rXKR9 due to its ability to form stable and monodisperse heterodimeric 1:1 complexes (*Figure 1—figure supplement 2*).

Following sample preparation, we collected large cryo-EM datasets of both complexes. The resulting maps, which extend to 3.66 Å for full-length and 4.3 Å for the cleaved protein, were of high quality and allowed in each case the interpretation with an atomic model (*Figure 2*, *Figure 2—figure supplements 1–4*, *Table 1*). The structure was initially built into the map of the full-length protein (*Figure 2A*). Although, due to the novelty of the protein fold, model building was challenging, the high quality of the map and the distribution of sidechain density of different volume have ultimately allowed the unambiguous interpretation of residues 1–66, 81–105, 119–344, and 365–373

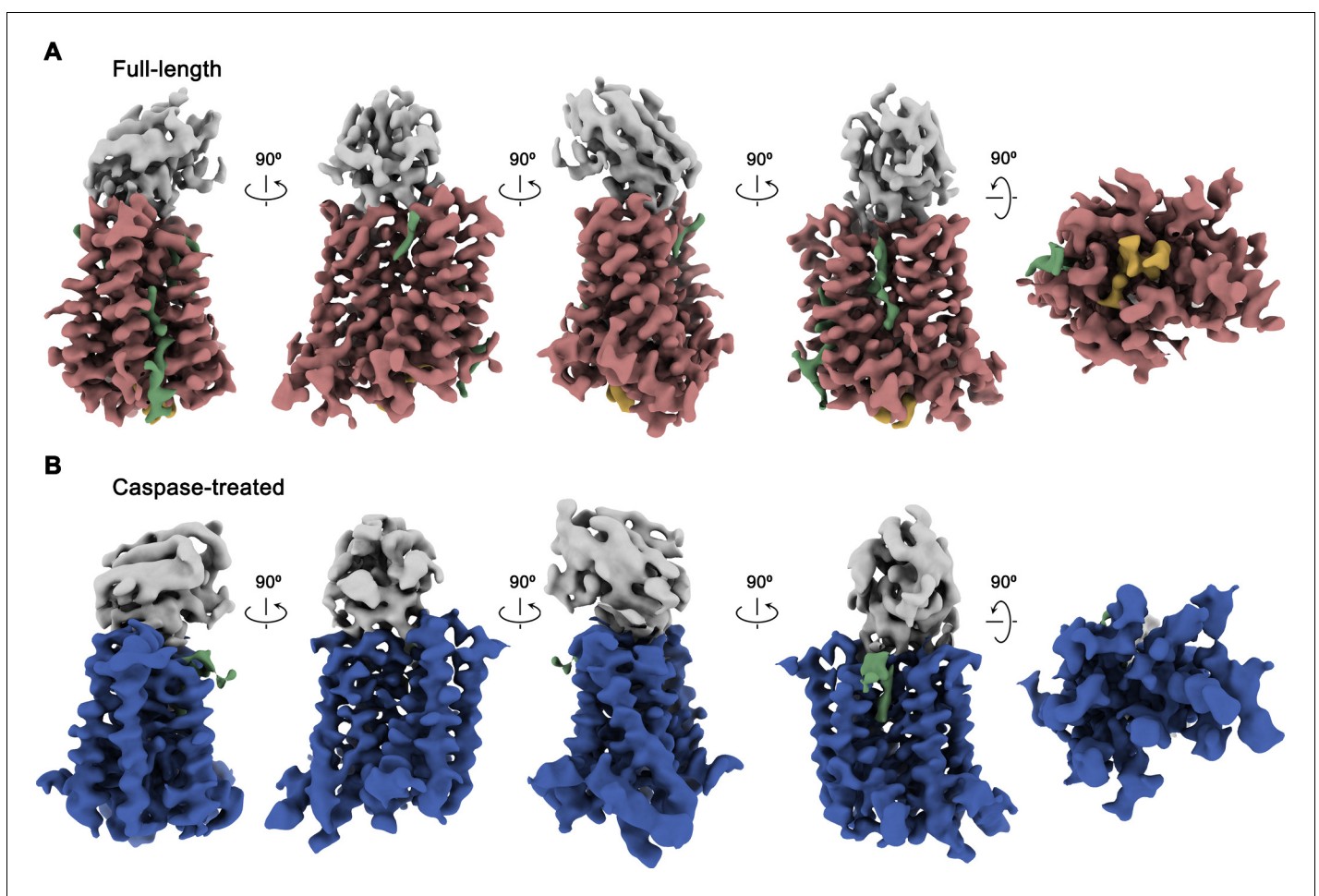

**Figure 2.** Cryo-EM maps of rXKR9-Sb1^XKR9 complexes. (**A**) Cryo-EM density of complexes of the full-length construct at 3.66 Å (contoured at 9.5 σ) and (**B**), the caspase-treated protein at 4.3 Å (contoured at 8.8 σ). The relationship between orientations is indicated. rXKR9 density is shown in unique colors, density of Sb1^XKR9 is shown in gray, residual density that can be attributed to bound lipids in green. Density of the C-terminus of full-length protein (**A**) is colored in orange.

The online version of this article includes the following figure supplement(s) for figure 2:

**Figure supplement 1.** Cryo-EM reconstruction of the full-length rXKR9-Sb1^XKR9 complex.

**Figure supplement 2.** Cryo-EM reconstruction of the caspase-treated rXKR9-Sb1^XKR9 complex.

**Figure supplement 3.** Model validation.

**Figure supplement 4.** Cryo-EM density.

**Table 1.** Cryo-EM data collection, refinement, and validation statistics.

| | Dataset 1 rXkr9 (EMDB-13155) (PDB 7P14) | Dataset 2 rXkr9 caspase −3 cleaved (EMDB-13157) (PDB 7P16) | Dataset 3 hXkr8+hBSG |
|---|---|---|---|
| Data collection and processing | | | |
| Microscope | FEI Titan Krios | FEI Titan Krios | Polara |
| Camera | Gatan K3 GIF | Gatan K3 GIF | Gatan K2 |
| Magnification | 130,000 | 130,000 | 130,000 |
| Voltage (kV) | 300 | 300 | 300 |
| Electron exposure (e⁻/Å²) | 70 | 70 | 35 |
| Defocus range (µm) | −1.0 to −2.4 | −1.0 to −2.4 | −0.8 to −2.5 |
| Pixel size * (Å) | 0.651 (0.3255) | 0.651 (0.3255) | 1.34 |
| Initial number of micrographs (no.) | 12,396 | 14,929 | 2212 |
| Initial particle images (no.) | 6,252,082 | 8,302,932 | 465,373 |
| Final particle images (no.) | 866,439 | 444,358 | 54,519 |
| Symmetry imposed | C1 | C1 | C1 |
| Map resolution (Å) FSC threshold | 3.66 0.143 | 4.3 0.143 | 15.3 0.143 |
| Map resolution range (Å) | 3.0–6.0 | 3.7–7.0 | |
| Refinement | | | |
| Model resolution (Å) FSC threshold | 3.7 0.5 | 4.3 0.5 | |
| Map sharpening b-factor (Å²) | −199.6 | −229.9 | |
| Model vs Map CC (mask) | 0.76 | 0.74 | |
| Model composition Non-hydrogen atoms Protein residues Ligand | 3796 456 PLC, P5S | 3621 445 PLC | |
| B factors (Å2) Protein Ligand | 53.43 47.62 | 68.75 49.33 | |
| R.m.s. deviations Bond lengths (Å) Bond angles (°) | 0.003 0.533 | 0.002 0.453 | |
| Validation MolProbity score Clashscore Poor rotamers (%) | 1.54 7.45 0.00 | 1.51 7.29 0.00 | |
| Ramachandran plot Favored (%) Allowed (%) Disallowed (%) | 97.3 2.7 0.00 | 97.48 2.52 0.00 | |

*Values in parentheses indicate the pixel size in super-resolution.

of rXKR9 encompassing all structured parts of the protein. In case of Sb1^XKR9, the density of the CDR regions in contact with the protein is well-resolved and reveals the interaction with its epitope, whereas the more peripheral residues are of lower resolution and were thus interpreted with a model that is based on a high-resolution structure of the constant region (*Figure 2—figure supplement 4B, D*). The structure of the cleaved rXKR9-Sb1^XKR9 complex was subsequently built using the full-length complex as template, which was comparably straightforward, since the bulk of the protein retained a similar conformation, and the density that was ascribed to the C-terminus in the data of the full-length protein was absent (*Figure 2B*).

The structures of the XKR9-sybody complexes are shown in *Figure 3*. Viewed perpendicular to the membrane, the entire complex is about 80 Å high, and measures 40×35 Å within the membrane plane. On the extracellular side, rXKR9 extends barely beyond the membrane boundary and does not contain any glycosylation sites. Sb1$^{XKR9}$ binds to this region burying 1466 Å$^2$ of the combined molecular surface. Sybody contacts are mainly formed by the long CDR3, which wraps around the loop connecting TM3–4 and wedges into the cleft towards the adjacent TM7–8 loop (*Figure 3—figure supplement 1*).

## XKR9 architecture

rXKR9 comprises eight membrane-spanning helices (TM1–8), which are connected by short loops on the extracellular and longer loops on the intracellular side (*Figure 4A–C*). Two particularly long intracellular regions harbor α-helices (IH1 and IH2) that are partly embedded in the membrane, whereas other segments are disordered and protrude into the cytoplasm. Although the number of transmembrane helices concurs with predictions by hydropathy analysis (*Krogh et al., 2001*), there is some

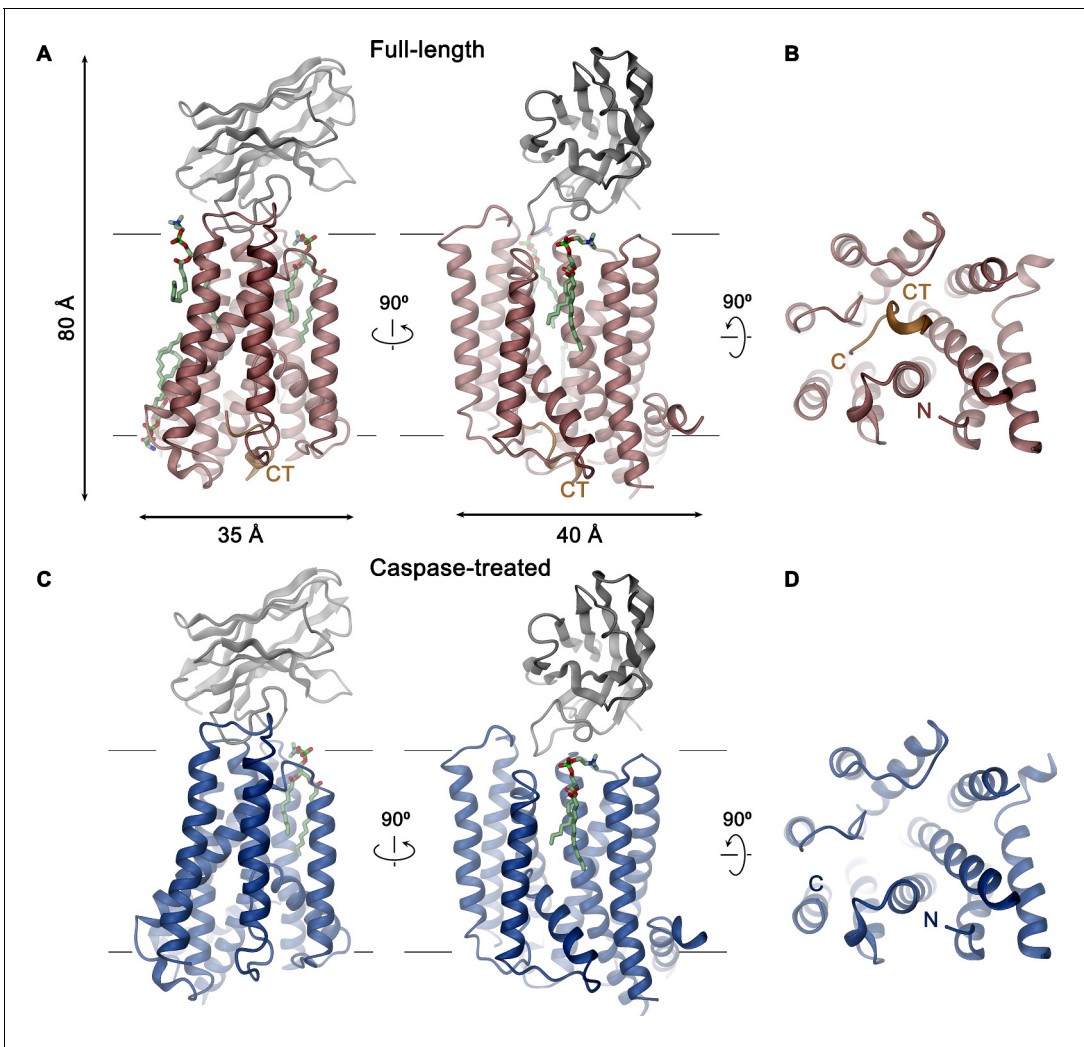

**Figure 3.** XKR9-SB1$^{XKR9}$ complex. Ribbon representation of Sb1$^{XKR9}$ complexes of (A, B) full-length and (C, D) caspase-treated rXKR9. The relationship between orientations is indicated. rXKR9 is shown in unique colors, Sb1$^{XKR9}$ in gray, and the C-terminus of the full-length protein (CT) that is cleaved upon caspase-3-treatment in orange. Bound lipids are shown in green as stick models. (A, C) View is from within the membrane, with membrane boundaries indicated. (B, D) View is from the cytoplasm.

The online version of this article includes the following figure supplement(s) for figure 3:

**Figure supplement 1.** rXKR9-Sb1$^{XKR9}$ interactions.

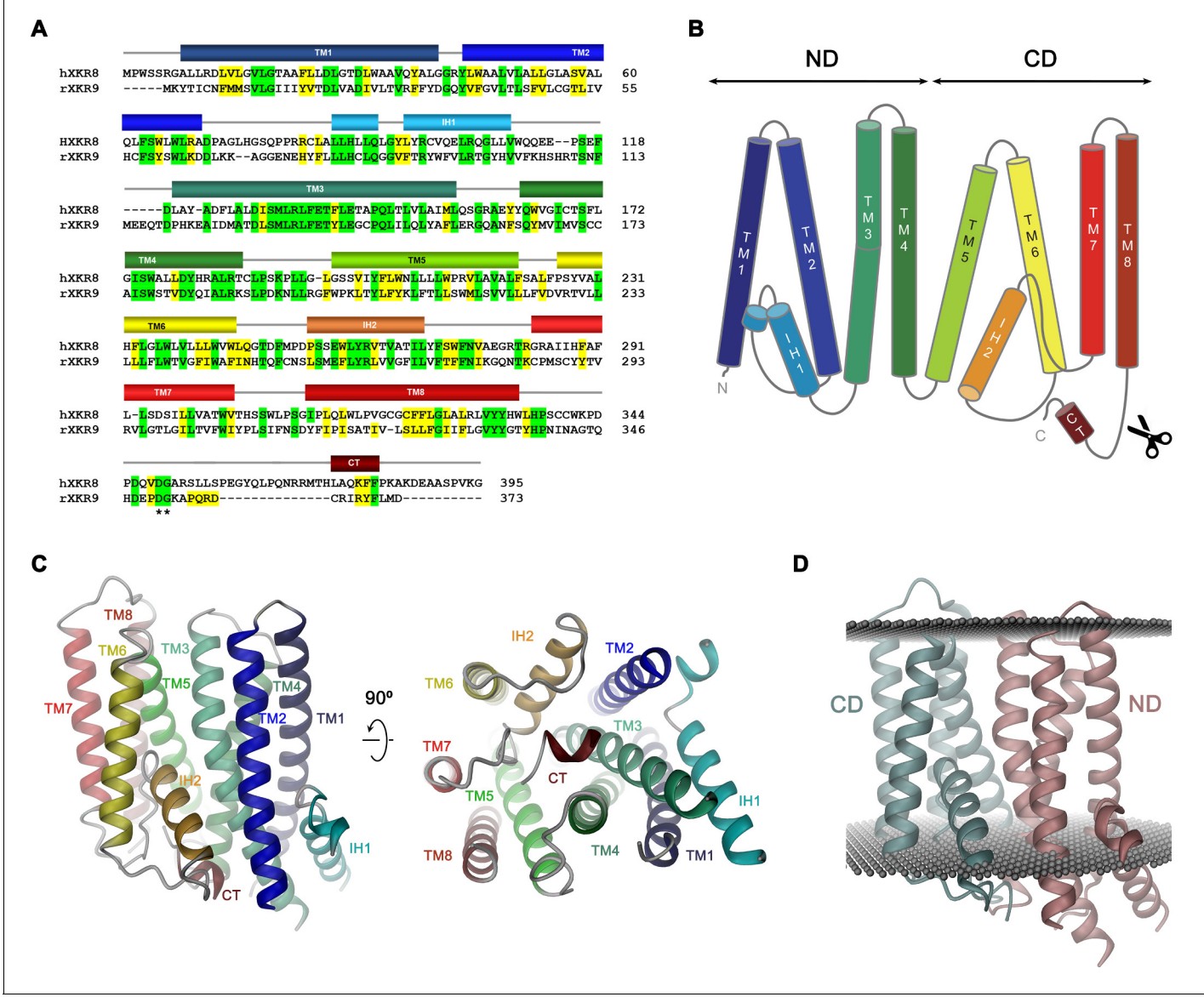

**Figure 4.** Sequence and topology. Sequence alignment of human XKR8 (hXKR8, NCBI: NP_060523.2) and rXKR9 (NCBI: NP_001012229.1), secondary structure elements are indicated. Identical residues are highlighted in green, similar residues in yellow. The caspase-3 cleavage site is marked by *. (B) Schematic topology of rXKR9. ND and CD refer to N-terminal and C-terminal domain, respectively. Scissors indicate location of caspase cleavage site. (C) Ribbon representation of the rXKR9 subunit. The relationship between orientations is indicated. Color code is as in (A) and (B). (D) Ribbon representation of rXKR9 with ND and CD shown in unique colors. Gray spheres indicate inner and outer boundaries of the hydrophobic core of the lipid bilayer.

The online version of this article includes the following figure supplement(s) for figure 4:

**Figure supplement 1.** Hydropathy analysis of the rXKR9 sequence.

discrepancy with respect to their location. Although seven of the helices were correctly predicted, the highly conserved TM3 was, due to its hydrophilic character, not assigned as membrane-spanning segment. Conversely, the sixth predicted segment forms the hydrophobic helix IH2 that only spans the inner membrane leaflet and is connected to TM7 by a re-entrant loop (*Figure 4—figure supplement 1*).

The rXKR9 structure consists of two consecutive domains (the N-terminal domain ND and C-terminal domain CD) with similar general organization containing four membrane-spanning helices each (*Figure 4B,D*). The first two helices of each repeat (i.e. TM1–TM2 and TM5–TM6) interact at

the extracellular side but are splayed open toward the inside to make space for the respective helical regions IH1 and IH2 that are contained within loops and inserted into the inner leaflet of the membrane (*Figure 4C*). The second pair of helices in each domain run parallel and make contacts throughout the membrane. IH1 is located at the periphery of the protein and engages in interactions with TM3. IH2, in contrast, is wedged between both domains to fill the intracellular part of a cavity that is formed at their interface (*Figure 4C,D*). The long hydrophilic transmembrane helix TM3 is kinked in the center of the membrane at a conserved PQL motive, which causes a bending away from the membrane normal on its intracellular part to maximize interactions with IH1 (*Figure 4C*). The general architecture of XKR9 closely resembles the recently reported structure of XKR8 (*Sakuragi et al., 2021*).

## Analysis of putative lipid pathways

Previous studies of the phospholipid scramblase nhTMEM16 have identified a membrane-spanning furrow as structural hallmark of the protein (*Brunner et al., 2014*). This polar furrow is of appropriate size for a phospholipid headgroup and might thus provide a favorable environment to accommodate this polar moiety on its way across the membrane. Consequently, we analyzed the rXKR9 structure for analogous features. Two furrow-like structures of appropriate size to harbor a lipid are found at the interface between the two domains of rXKR9 on both sides of the molecule and an additional structural feature of similar shape is contained within the C-terminal domain (*Figure 5A*). On one side of the domain interface, a cavity formed by the helices TM2, 3, 5, and 6 (termed C1) is on its intracellular part filled with the helix IH2 (*Figure 5A,B*). Its appropriate dimensions are underlined by a bound lipid molecule observed in our structure (*Figure 5B*, *Figure 5—figure supplement 1A,B*). This lipid is located at the expected height of the outer membrane leaflet and its fatty acid chains are surrounded by hydrophobic and aromatic residues. On the opposite side of the molecule, a similar cavity (C2) is formed by TM3, 4, 5, and 8, which is filled toward the intracellular side by the tilted α-helix TM5 (*Figure 5A,C*). As the cavity C1, also C2 contains residual density indicating the presence of bound phospholipids (*Figure 5—figure supplement 1A,C,D*). On its intracellular side, outside of the hydrophobic membrane core, this region contains a cluster of basic amino acids, which might interact with the headgroups of negatively charged lipids as evidenced by close-by lipid density (*Figure 5—figure supplements 1D* and *2C*). The third cavity (C3) contained within the C-terminal domain is formed by helices TM5, 6, and 7 (*Figure 5A,D*). Although we also find residual density contained in this cavity, it does not resemble lipid molecules (*Figure 2A*). In all three cases, the generally hydrophobic character of the described cavities renders their suitability as a lipid translocation path ambiguous (*Figure 5A–D*, *Figure 5—figure supplement 1B–D*).

A second hallmark of a lipid scramblase relates to polar residues that are embedded within the membrane-inserted part of the protein, which in its active conformation, would be exposed to the bilayer to lower the energy barrier for lipid flip-flop (*Bethel and Grabe, 2016*; *Brunner et al., 2014*; *Jiang et al., 2017*; *Lee et al., 2018*; *Stansfeld et al., 2015*). Although the transmembrane helices of rXKR9 contain several polar and charged residues, which have precluded the assignment of TM3 as membrane-spanning region during hydropathy analysis (*Figure 4A*, *Figure 4—figure supplement 1*), the observed protein conformation buries these residues between the tightly packed transmembrane segments. Due to the long-range nature of coulombic interactions and the low dielectrics of the bilayer, these residues might still influence the electrostatics on the surface of the close-by cavities C1 and C2 and thus their ability to interact with polar compounds (*Figure 5—figure supplement 2*). A region with high density of acidic residues is positioned between interacting helices of the N-terminal domain (*Figure 5E,F*). These residues, which are predominantly located on TM1 and 3, are strongly conserved throughout the family and thus appear to be a characteristic feature of XKR proteins (*Figure 5F,G*). Occluded within the core of the N-terminal domain, the vicinity of several acidic residues distantly resembles a divalent cation binding site, although the structure does not provide evidence of bound cations (*Figure 5F*). Conversely, a cluster of polar and basic residues is located at the C-terminal domain and render the electrostatics positive (*Figure 5H–J*). In contrast to the N-terminal domain, these residues are less conserved (*Figure 5J*) and one buried arginine (Arg 294) appears to interact with the C-terminus of the partly inserted helix IH2, suggesting that it would serve a role in the stabilization of the structure (*Figure 5H,I*).

In summary, while the presence of charged residues located within the membrane domain, some of which are strongly conserved, emphasize their importance for protein stability, their role for lipid

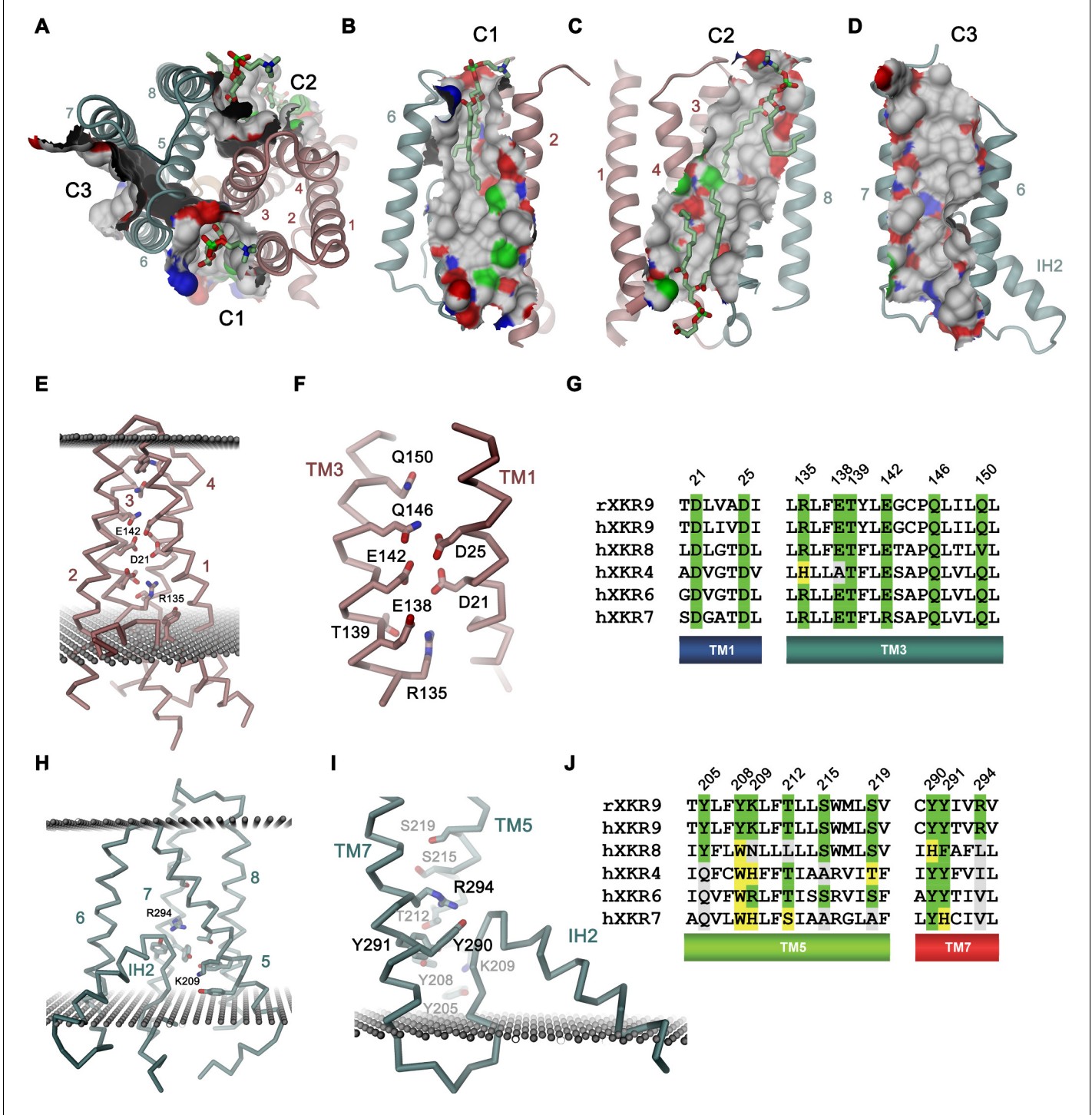

**Figure 5.** Features of the rXKR9 structure of potential functional relevance. (A–D) Cavities on the surface of the rXKR9 structure that are suitable for lipid interactions. Segments of the protein are displayed as ribbon. Sections of the molecular surface are shown and colored according to the contacting atoms (carbon: gray, oxygen: red, nitrogen: blue, sulfur: green). Transmembrane helices are numbered and cavities are labeled (C1–C3). (A) View from the extracellular side. (B) Cavities C1, (C), C2, and (D), C3 viewed from within the membrane. (A–C) Bound lipids are shown as sticks. (E) Structure of polar residues buried in the ND. (F) Closeup of the membrane-inserted parts of TM1 and 3 displaying the position of conserved polar and charged residues. (G) Sequence alignment of the regions displayed in (F) in rXKR9 and selected human paralogs. (H) Structure of polar residues buried in the CD. (I) Closeup of the membrane inserted region encompassing TM5, TM7 and IH2 displaying the position of polar and charged residues. (J) Sequence alignment of the regions displayed in (H) in rXKR9 and selected human paralogs. (G, J) Alignment of selected regions of rXKR9 (NCBI: NP_001012229.1) with hXKR9 (NCBI: NP_001011720.1), hXKR8 (NCBI: NP_060523.2), hXKR4 (NCBI: NP_443130.1), hXKR6 (NCBI: NP_775954.2), and

*Figure 5 continued on next page*

*Figure 5 continued*

hXKR7 (NCBI: NP_001011718.1). Secondary structure elements are indicated. Identical residues are highlighted in green, similar residues in yellow. (E, H, I) Boundaries of the hydrophobic core of the bilayer are indicated as spheres. (C, E, H, I) The protein is displayed as Cα-trace with selected residues shown as sticks. Secondary structure elements and selected residues are labeled. (A–F, H, I) Coloring of the protein is as in *Figure 4D*, ND in red and CD in cyan.

The online version of this article includes the following figure supplement(s) for figure 5:

**Figure supplement 1.** Lipid density.

**Figure supplement 2.** Surface electrostatics.

movement, and the question whether some of them would change their exposure during activation is still unclear.

## Caspase cleavage of the C-terminus

Activation of apoptotic scramblases proceeds after cleavage of a C-terminal peptide by caspase-3 at a specific recognition site (*Figure 4A*). In case of rXKR9, part of the cleavable peptide, which is only 16 residues long, binds to an intracellular site that is located in a pocket at the interface between the two domains formed by residues of TM3-5, 8, IH2 and the connecting loop regions (*Figure 6A, B*). In the structure of full-length rXKR9, part of the inhibitory peptide is well-defined, whereas the

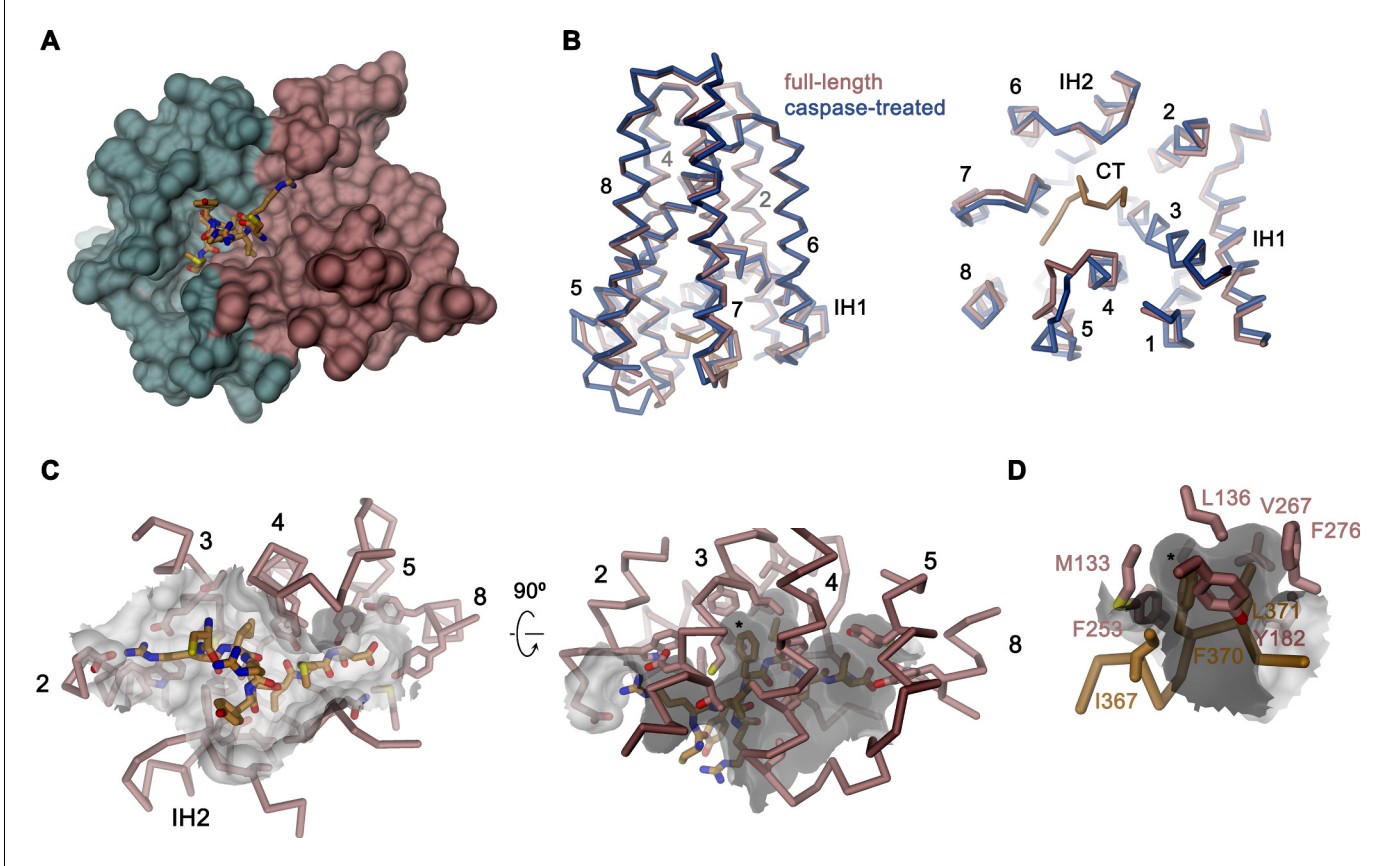

**Figure 6.** Binding site of the C-terminus. (A) Binding of the C-terminus of rXKR9 to a cleft of the protein. rXKR9 excluding its C-terminus is shown as surface representation and colored as in *Figure 4D*. The C-terminal peptide (CT) defined in the structure is shown as stick model. (B) Superposition of the full-length and caspase-cleaved structures of rXKR9. Left, view from within the membrane. Right, view from the cytoplasm. (C) Interactions of the CT with residues lining its binding site in rXKR9. The relationship between views is indicated. (left, view from the cytoplasm, right, view from within the membrane). (D) Blow-up of hydrophobic residues of the CT with its binding site. (C, D) * marks position of F370. (B–D) protein is shown as Cα-trace with selected residues shown as sticks. Sections of the molecular surface are displayed. Membrane-spanning helices are indicated by numbers, IH1 and IH2 and CT and selected residues are labeled explicitly.

connecting loop harboring the cleavage site appears unstructured (*Figure 2A*, *Figure 2—figure supplement 4*). Features in the density suggest the binding of eight residues at the very C-terminus of the protein. These fold into a single turn of an α-helix followed by an extended region that form complementary interactions with the protein burying 1698 $\text{Å}^2$ of the combined molecular surface (*Figures 3B* and *6A*). In our model, the C-terminus fills the binding pocket to either stabilize the protein in its inactive form or, alternatively, shield the residues lining the pocket from other interactions. The binding is predominantly hydrophobic at its core but involves coulombic interactions at the periphery (*Figure 6C*). An isoleucine (Ile 367), a phenylalanine (Phe 370), and a leucine (Leu 371) of the C-terminus are buried in a hydrophobic pocket of the protein that is lined by aliphatic and aromatic residues (*Figure 6D*). In the data of the cleaved rXKR9, the density of the C-terminus is absent, exposing the hydrophobic cleft to the cytoplasm (*Figure 2*). Apart from smaller changes in the intracellular loop regions, the truncated protein has not changed its conformation, which is illustrated in the low RMSD of 0.72 Å between the two structures (*Figure 6B*). However, at this stage we do not want to exclude the possibility of larger conformational changes during activation as the observed conformation might be either be stabilized by the bound sybody or the absence of an endogenous factor that has also prohibited lipid scrambling in reconstituted samples (*Figure 1C*).

## Discussion

By determining the structure of full-length rXKR9 and its caspase-cleaved form by cryo-EM, our study has revealed the architecture of the XKR family and provided a first glimpse on its activation. With a size of only 43 kDa and not containing pronounced structural features outside of the membrane, rXKR9 on its own is too small for cryo-EM analysis. However, the addition of a sybody increasing the complex size to 60 kDa was sufficient to obtain high-quality data and thus could provide a suitable general strategy for the structure determination of small and asymmetric membrane proteins. The rXKR9 structures show a novel protein fold, consisting of two structurally related repeats of four transmembrane helices each, that is presumably shared by the entire family irrespective of the function of its members, which is in many cases still elusive.

Three members of the XKR family, namely the proteins XKR8, 9, and 4, were previously identified as essential constituents of apoptotic lipid scramblases, which are activated by proteolysis (*Suzuki et al., 2014*). Caspases directly act on the protein by targeting a single cleavage site located after the last transmembrane helix TM8 (*Figure 4A*). Our structures illustrate the consequence of this cleavage where the C-terminus, which binds to a hydrophobic pocket facing the cytoplasm in the full-length protein, is released to free the access to this site (*Figures 4B,D* and *6*). The 16-residue-long peptide thus acts as inhibitor of protein function, whose dissociation from its binding site is a requirement for activation. This could proceed either by lowering its local concentration following proteolysis or by covalent modifications of the peptide as recently suggested for XKR8, which is also activated by phosphorylation of specific positions of the C-terminus (*Sakuragi et al., 2019*).

Although in our structure of the cleaved rXKR9, we do not observe any larger rearrangements of the protein as a consequence of proteolysis (*Figure 6B*), such conformational changes could be prevented by the bound sybody or the lack of accessory factors, which might lock the protein in its observed conformation. Although previous experimental data suggested that XKR proteins themselves form the catalytic unit promoting scrambling activity (*Suzuki et al., 2013*; *Suzuki et al., 2014*), our attempts to observe such activity for rXKR9 in vitro after purification and reconstitution into liposomes did not provide evidence for lipid flip-flop (*Figure 1D*). This does not exclude that rXKR9 would itself act as a scramblase, but it suggests that there might be additional components required for its activity. In case of its paralog XKR8, this could be the interaction with the β-subunit basigin (*Figure 1E,F*) and the proposed dimerization of the complex upon activation (*Suzuki et al., 2016*). Pairing in response to caspase cleavage was recently also reported for the protein XKR4, which additionally requires the binding of an endogenous peptide for activation (*Maruoka et al., 2021*). For rXKR9, we did neither observe an interaction with basigin, nor did we find any indication for the change of the oligomeric state following caspase cleavage (*Figure 1B,D*), leaving the possibility that the protein might require an unknown subunit for activation and lipid transport.

Previous studies on fungal homologs of the TMEM16 family have defined features that underlie their function as lipid scramblases. These include a protein architecture that distorts the membrane and the presence of a hydrophilic membrane-spanning furrow that is hidden in the closed

conformation and that becomes exposed in the activated conformation of the protein (*Bethel and Grabe, 2016*; *Brunner et al., 2014*; *Falzone et al., 2019*; *Kalienkova et al., 2019*). Both factors appear sufficient to lower the energetic barrier for lipid flip-flop, which might proceed in the immediate vicinity of the polar furrow and even at some distance from the protein (*Malvezzi et al., 2018*). No such obvious features for scramblase function were found in the rXKR9 structures (*Figure 2—figure supplement 4A,C*). The micelle surrounding the protein does not show clear evidence for distortion, which in fungal TMEM16 scramblases was already manifested in structures determined in detergents (*Kalienkova et al., 2019*). Moreover, appropriate cavities located at the surface of the protein, which contain residual densities of bound lipids, are generally hydrophobic and thus do no offer obvious interactions with the lipid headgroups on their way across the membrane (*Figures 2* and *5A–D*, *Figure 5—figure supplements 1* and *2*). Polar amino acids on the surface of rXKR9 are generally found in the headgroup region, including a cluster of positively charged residues located at the intracellular side of one cavity, which might facilitate the interaction with negatively charged lipids (*Figure 5—figure supplements 1D* and *2C*). A remarkable feature of rXKR9 concerns a cluster of acidic residues buried in the protein, which is strongly conserved within family members that were assigned as lipid scramblases, with TM3 being unusually hydrophilic for a transmembrane helix (*Figure 5E–G*). The role of these residues for protein function is currently unknown, and it is conceivable that they might contribute to lipid scrambling in an active state of the protein, which is likely not displayed in any of our structures. Due to the low dielectric environment of the membrane, charged residues might influence the interaction with polar lipid headgroups even in case they would remain buried after activation (*Figure 5—figure supplement 2*). In light of the lack of structural evidence for its scramblase function, it is noteworthy that opsins were demonstrated to function as lipid scramblases without containing obvious structural features (*Menon et al., 2011*; *Morra et al., 2018*) and that the hallmarks described for the fungal scramblases nhTMEM16 (*Brunner et al., 2014*), afTMEM16 (*Falzone et al., 2019*), or TMEM16K (*Bushell et al., 2019*) are not displayed in any of the current structures of TMEM16F, whose function as lipid scramblase has been demonstrated in vivo and in vitro (*Alvadia et al., 2019*; *Suzuki et al., 2010*).

In summary, our study has defined the architecture of a protein that forms a critical part of the machinery leading to the exposure of PS during programmed cell death, which is an essential step leading to the clearance of apoptotic cells (*Segawa and Nagata, 2015*). By defining the interaction of the C-terminus, which is removed upon proteolysis, it has also provided initial insight into the activation of family members by caspases. However, as we did not observe activity of rXKR9 upon reconstitution and its structures do not provide immediate evidence for a potential scrambling mechanism, we envisage different scenarios for activation and lipid transport. These could either include a conformational change that would expose polar patches that are buried in the inactive protein to facilitate lipid flip-flop, dimerization of the protein as suggested for XKR8 and XKR4 (*Maruoka et al., 2021*; *Suzuki et al., 2016*), or binding of an extrinsic interaction partner that might contribute part or all of the site responsible for lipid flip-flop. These mechanisms could also work in concert to promote the activation of an apoptotic scramblase. Collectively, our work provides a foundation for future experiments that will have to clarify whether and how the members of this important protein family catalyze lipid translocation between membrane leaflets and identify interaction partners that may play a critical role in their function.

## Materials and methods

### Key resources table

| Reagent type (species) or resource | Designation | Source or reference | Identifiers | Additional information |
|---|---|---|---|---|
| Cell line (human) | HEK239T | ATCC | CRL-3216 | RRID:CVCL_0063 |
| Cell line (human) | HEK293S GnTI⁻ | ATCC | CRL-3022 | RRID:CVCL_A785 |
| Cell line (human) | mTMEM16F HEK293T | *Alvadia et al., 2019* | NA | |

*Continued on next page*

*Continued*

| Reagent type (species) or resource | Designation | Source or reference | Identifiers | Additional information |
|---|---|---|---|---|
| Chemical compound, drug | 18:1 06:0 NBD-PE | Avanti Polar Lipids | 810155C | |
| Chemical compound, drug | 18:1 06:0 NBD-PS | Avanti Polar Lipids | 810194C | |
| Chemical compound, drug | 1-palmitoyl-2-oleoyl-sn-glycero-3-phospho-(1'rac-glycerol) (18:1 06:0 POPG) | Avanti Polar Lipids | 840457C | |
| Chemical compound, drug | 1-palmitoyl-2-oleoyl-sn-glycero-3-phosphocholine (18:1 06:0 POPC) | Avanti Polar Lipids | 850457C | |
| Chemical compound, drug | 1-palmitoyl-2-oleoyl-sn-glycero-3-phosphoeth anolamine (18:1 06:0 POPE) | Avanti Polar Lipids | 850757C | |
| Chemical compound, drug | Benzamidine | Sigma | B6506 | |
| Chemical compound, drug | Biotin | Sigma | B4501 | |
| Chemical compound, drug | Calcium nitrate tetrahydrate | Sigma | C4955 | |
| Chemical compound, drug | CHAPS | Sigma | C3023 | |
| Chemical compound, drug | Chloramphenicol | Sigma | C1919 | |
| Chemical compound, drug | Chloroform | Fluka | 25690 | |
| Chemical compound, drug | Cholesterol | Sigma | C3045 | |
| Chemical compound, drug | cOmplete, EDTA-free Protease Inhibitor Cocktail | Roche | 5056489001 | |
| Chemical compound, drug | D-Desthiobiotin | Sigma | D1411 | |
| Chemical compound, drug | DDM | Anatrace | D310S | |
| Chemical compound, drug | Diethyl ether | Sigma | 296082 | |
| Chemical compound, drug | Digitonin High Purity | Merck Millipore | 300410 | |

*Continued on next page*

*Continued*

| Reagent type (species) or resource | Designation | Source or reference | Identifiers | Additional information |
|---|---|---|---|---|
| Chemical compound, drug | DNase I | AppliChem | A3778 | |
| Chemical compound, drug | DTT | ThermoFisher Scientific | R0861 | |
| Chemical compound, drug | Dulbecco's modified Eagle's medium - high glucose | Gibco, Thermo Fisher Scientific | 41966029 | |
| Chemical compound, drug | Etyleneglycol - bis (2 - aminoethylether) - N,N,N',N'- tetraacetic acid | Sigma | E3889 | |
| Chemical compound, drug | Fetal bovine serum | Sigma | F7524 | |
| Chemical compound, drug | GDN | Anatrace | GDN101 | |
| Chemical compound, drug | Glycerol 99% | Sigma | G7757 | |
| Chemical compound, drug | HCl | Merck Millipore | 1.00319.1000 | |
| Chemical compound, drug | HEPES | Sigma | H3375 | |
| Chemical compound, drug | HyClone HyCell TransFx-H medium | Cytiva | SH30939.02 | |
| Chemical compound, drug | Imidazole | Roth | X998.4 | |
| Chemical compound, drug | Kolliphor P188 | Sigma | K4894 | |
| Chemical compound, drug | L-(+)-arabinose | Sigma | A3256 | |
| Chemical compound, drug | Lauryl-maltose-neopentyl glycol (LMNG) | Anatrace | NG310 | |
| Chemical compound, drug | Leupeptin | AppliChem | A2183 | |
| Chemical compound, drug | L-glutamine | Sigma | G7513 | |
| Chemical compound, drug | Penicillin–streptomycin | Sigma | P0781 | |
| Chemical compound, drug | Pepstatin | AppliChem | A2205 | |

*Continued on next page*

*Continued*

| Reagent type (species) or resource | Designation | Source or reference | Identifiers | Additional information |
|---|---|---|---|---|
| Chemical compound, drug | Phosphate buffered saline | Sigma | D8537 | |
| Chemical compound, drug | PMSF | Sigma | P7626 | |
| Chemical compound, drug | Polyethyleneimine MAX 40 kDa | PolySciences Inc | 24765–1 | |
| Chemical compound, drug | Potassium chloride | Sigma | 746346 | |
| Chemical compound, drug | Sodium chloride | Sigma | 71380 | |
| Chemical compound, drug | Sodium dithionite (Sodium hydrosulfite) | Sigma | 157953 | |
| Chemical compound, drug | Soybean Polar Lipids | Avanti Polar Lipids | 541602C | |
| Chemical compound, drug | TCEP | Hampton | HR2-651 | |
| Chemical compound, drug | Terrific broth | Sigma | T9179 | |
| Chemical compound, drug | Tetracycline hydrochloride | Sigma | T7660 | |
| Chemical compound, drug | Tris | AppliChem | A1379 | |
| Chemical compound, drug | Valproic acid | Sigma | P4543 | |
| Commercial assay or kit | 4–20% Mini-PROTEAN TGX Precast Protein Gels, 15-well, 15 µl | BioRad Laboratories | 4561096DC | |
| Commercial assay or kit | Amicon Ultra-4 Centrifugal Filters Ultracel 10K, 4 ml | Merck Millipore | UFC801096 | |
| Commercial assay or kit | Amicon Ultra-4 Centrifugal Filters Ultracel 50K, 4 ml | Merck Millipore | UFC805096 | |
| Commercial assay or kit | Amicon Ultra-4 Centrifugal Filters Ultracel 100K, 4 ml | Merck Millipore | UFC810024 | |
| Commercial assay or kit | Biobeads SM-2 adsorbents | BioRad Laboratories | 152–3920 | |
| Commercial assay or kit | EZ-link NHS-PEG4-biotin | ThermoFisher Scientific | A39259 | |
| Commercial assay or kit | MiniExtruder | Avanti Polar Lipids | 610023 | |

*Continued on next page*

*Continued*

| Reagent type (species) or resource | Designation | Source or reference | Identifiers | Additional information |
|---|---|---|---|---|
| Commercial assay or kit | Ni-NTA resin | ABT Agarose Bead Technologies | 6BCL-NTANi-X | |
| Commercial assay or kit | Nunc 384-well plate MaxiSorp | ThermoFisher Scientific | 460372 | |
| Commercial assay or kit | PC membrane 0.2 μm | Avanti Polar Lipids | 610006 | |
| Commercial assay or kit | PD-10 desalting column | Sigma | GE17-0851-01 | |
| Commercial assay or kit | Pierce Streptavidin Plus UltraLink Resin | ThermoFisher Scientific | 53117 | |
| Commercial assay or kit | QuantiFoil R1.2/1.3 Au 200 mesh | Electron Microscopy Sciences | Q2100AR1.3 | |
| Commercial assay or kit | Silver Stain Plus | BioRad Laboratories | 161–0449 | |
| Commercial assay or kit | SRT-10C SEC 100 | Sepax Technologies | 239100–10030 | |
| Commercial assay or kit | Strep-Tactin Superflow high capacity 50% suspension | IBA LifeSciences | 2-1208-010 | |
| Commercial assay or kit | Superdex 200 10/300 GL | Cytiva | 17517501 | |
| Commercial assay or kit | Superdex 200 Increase 3.2/300 | Cytiva | 28990946 | |
| Commercial assay or kit | Superdex 200 Increase 5/150 GL | Cytiva | 28990945 | |
| Commercial assay or kit | Superdex 75 10/300 GL | Cytiva | 17517401 | |
| Commercial assay or kit | Superose 6 10/300 GL | Cytiva | 17517201 | |
| Commercial assay or kit | Superose 6 Increase 5/150 | Cytiva | 29091597 | |
| Commercial assay or kit | Ultrafree MC GV 0.22 μm centrifugal filter | Merck Millipore | UFC30GVNB | |
| Commercial assay or kit | Ultrafree - MC -HV, DURAPORE PVDF 0.1 μm | Merck Millipore | UFC30VV00 | |
| Other | BioQuantum Energy Filter | Gatan | NA | |
| Other | Fluoromax spectrometer | Horiba | NA | |
| Other | HPL6 | Maximator | NA | |
| Other | K2 Summit Direct Detector | Gatan | NA | |
| Other | K3 Summit Direct Detector | Gatan | NA | |
| Other | qPCR machine Mx3005p | Agilent | NA | |

*Continued*

| Reagent type (species) or resource | Designation | Source or reference | Identifiers | Additional information |
|---|---|---|---|---|
| Other | Sybody library | Generous gift from the Seeger laboratory | NA | |
| Other | Tecnai G$^2$ Polara | ThermoFisher Scientific | NA | |
| Other | Titan Krios G3i | ThermoFisher Scientific | NA | |
| Other | UT-rEX Refractometer | Wyatt Technology | NA | |
| Other | Viber Fusion FX7 imaging system | Witec | NA | |
| Other | Vitrobot Mark IV | ThermoFisher Scientific | NA | |
| Other | μDAWN MALS Detector | Wyatt Technology | NA | |
| Recombinant DNA reagent | human Basigin open reading frame | GenScript | HUMBSG | |
| Recombinant DNA reagent | human Xkr8 open reading frame | GenScript | NM_018053 | |
| Recombinant DNA reagent | Mammalian expression vector with C-terminal 3C cleavage site, GFP-tag and myc-tag | Dutzler laboratory | NA | |
| Recombinant DNA reagent | Mammalian expression vector with N-terminal streptavidin binding peptide, myc-tag and 3C cleavage site | Dutzler laboratory | NA | |
| Recombinant DNA reagent | Mammalian expression vector with N-terminal streptavidin binding peptide, myc-tag, Venus-tag and 3C cleavage site | Dutzler laboratory | NA | |
| Recombinant DNA reagent | Mouse mTMEM16F open reading frame | Dharmacon - Horizon Discovery | GenBank#BC060732 | |
| Recombinant DNA reagent | pSb_init | Generous gift from the Seeger laboratory | RRID:addgene_110100 | |
| Recombinant DNA reagent | rat basigin open reading frame | GenScript | NM_012783 | |
| Recombinant DNA reagent | rat XKR9 open reading frame | GenScript | NM_001012229 | |
| Peptide, recombinant protein | HRV 3C protease | Expressed (pET_3C) and purified in Dutzler laboratory | NA | |

*Continued on next page*

*Continued*

| Reagent type (species) or resource | Designation | Source or reference | Identifiers | Additional information |
|---|---|---|---|---|
| Peptide, recombinant protein | Human Caspase-3, recombinant | BioVision | 1083 | |
| Peptide, recombinant protein | Streptavidin | Expressed and purified in Dutzler laboratory | NA | |
| Software, algorithm | 3DFSC | *Tan et al., 2017* | https://3dfsc.salk.edu/ | |
| Software, algorithm | ASTRA7.2 | Wyatt Technology | https://www.wyatt.com/products/software/astra.html | RRID:SCR_016255 |
| Software, algorithm | Chimera v.1.15 | *Pettersen et al., 2004* | https://www.cgl.ucsf.edu/chimera/ | RRID:SCR_004097 |
| Software, algorithm | ChimeraX v.1.1.1 | *Pettersen et al., 2021* | https://www.rbvi.ucsf.edu/chimerax/ | RRID:SCR_015872 |
| Software, algorithm | Coot v.0.9.4 | *Emsley and Cowtan, 2004* | https://www2.mrc-lmb.cam.ac.uk/personal/pemsley/coot/ | RRID:SCR_014222 |
| Software, algorithm | cryoSPARC v.3.0.1/v.3.2.0 | Structura Biotechnology Inc | https://cryosparc.com/ | RRID:SCR_016501 |
| Software, algorithm | DINO | | http://www.dino3d.org | RRID:SCR_013497 |
| Software, algorithm | EPU2.9 | ThermoFisher Scientific | NA | |
| Software, algorithm | Codon optimization tool | Integrated DNA technologies | https://eu.idtdna.com/pages/tools/codon-optimization-tool | |
| Software, algorithm | Phenix | Liebschner et al., 2019 | https://www.phenix-online.org/ | RRID:SCR_014224 |
| Software, algorithm | RELION 3.0.7 | *Zivanov et al., 2018* | https://www3.mrc-lmb.cam.ac.uk/relion/ | RRID:SCR_016274 |
| Software, algorithm | SerialEM3.7 | Mastronarde, 2005 | https://bio3d.colorado.edu/SerialEM/ | |
| Software, algorithm | WEBMAXC calculator | *Bers et al., 2010* | https://somapp.ucdmc.ucdavis.edu/pharmacology/bers/maxchelator/webmaxc/webmaxcS.htm | RRID:SCR_018807 |
| Strain | *E. coli* MC1061 | ThermoFisher Scientific | C66303 | |

## Cell culture

HEK293S GnTI⁻ and HEK293T cells were obtained from ATCC. mTMEM16F HEK293T cells were previously generated in our lab using the Flp-In T-REx system (*Alvadia et al., 2019*). All cell lines were tested negative for mycoplasma. HEK293S GnTI⁻ and mTMEM16F HEK293T cells were adapted to suspension cultures and grown in HyCell HyClone Trans Fx-H media supplemented with 1% FBS, 4 mM L-glutamine, 100 U/ml penicillin–streptomycin, and 1.5 g/l Kolliphor-184 at 37°C and 5% $CO_2$. Adherent HEK293T cells were grown in DMEM, supplemented with 10% FBS and 100 U/ml penicillin/streptomycin.

## Construct preparation

Genes used in this study were codon-optimized for mammalian expression (IDT codon optimization tool), synthesized by GenScript, and cloned into pcDNA 3.1 (pcDx) expression vectors by FX-cloning (*Geertsma and Dutzler, 2011*). rXKR9 (molecular weight (MW) 43.6 kDa) and hXKR8 (MW 51.5 kDa) were each cloned into two different pcDx vectors, one containing an N-terminal streptavidin binding peptide (SBP), a myc-tag, and a human rhinovirus (HRV) 3C cleavage site (pcDxNSM3), and the other harboring an additional Venus YFP tag in front of the 3C cleavage site (pcDxNSMV3). For co-purification, rat and human basigin (respective MWs of 59.0 and 58.8 kDa) were cloned into a pcDx vector adding a 3C cleavage site, GFP, and a myc-tag (pcDxC3GM) to the C-terminus of the expressed protein.

The sybody sequences were cloned into the FX-compatible, chloramphenicol-resistant, and arabinose-inducible pSb_init vector (*Zimmermann et al., 2020*), containing an N-terminal pelB leader sequence and a C-terminal hexa-histidine tag.

## Protein expression and purification

For transient transfection of rXKR9, purified plasmid DNA was mixed with PEI MAX in a 1:2.5 ratio, diluted in non-supplemented DMEM medium to 0.01 mg DNA per ml, and incubated for 15 min at room temperature prior to addition to the cells. Valproic acid (3.5 mM) was added immediately after transfection. Cells were harvested 48 hr post-transfection, washed in PBS, and pellets were frozen in liquid $N_2$ and stored at $-80°C$ until further use.

Frozen pellets were thawed on ice, re-suspended in lysis buffer (25 ml buffer per 1 l of suspension culture), containing 25 mM HEPES pH 7.5, 200 mM NaCl and either 1% LMNG in case of rXKR9 or 2% GDN in case of hXKR8-basigin and protease inhibitors (0.1 μM PMSF, 10 μM leupeptin, 1 mM benzamidine, 1 μM pepstatin). All purification steps were carried out at 4°C. Membrane proteins were solubilized for 2 hr, while gently mixing. Insoluble fractions were removed by ultracentrifugation at 85,000 g for 30 min and supernatant was incubated with Streptactin superflow resin (1 ml 50% slurry for 1 l of suspension cell culture) for 2.5 hr under gentle agitation. The resin was washed with 50 column volumes (CV) of size-exclusion chromatography (SEC) buffer (25 mM HEPES pH 7.5, 200 mM NaCl, and either 0.01% LMNG or 0.03% GDN). The protein was eluted with 5 CV SEC buffer, supplemented with 10 mM desthiobiotin, and incubated at a 1:1 (wt/wt) ratio with HRV 3C protease for 45 min. The cleaved protein was concentrated using a 50 kDa cut-off concentrator for rXKR9 and a 100 kDa cut-off concentrator for hXKR8-basigin. After filtration (0.22 μm filter) proteins were either injected onto a Superdex 200 10/300 GL column (rXKR9) or a Superose 6 10/300 GL clumn (hXKR8-basigin),equilibrated in respective SEC buffers. Peak fractions were collected and concentrated as described above.

The sybody Sb1$^{XKR9}$ (MW 17.0 kDa) was transformed into *E. coli* MC1061. Bacteria were grown in 1.2 l Terrific Broth media supplemented with 25 μg/ml chloramphenicol at 37°C. At $OD_{600} = 0.5$, the temperature was decreased to 22°C for 1.5 hr and expression was induced with 0.02% ʟ-arabinose for 16–18 hr. Bacterial cultures were harvested at 4,000 g for 20 min at 4°C, and pellets were frozen in liquid $N_2$ and stored at $-80°C$. Frozen pellets were thawed on ice, re-suspended in lysis buffer TBS (36.5 mM Tris–HCl pH 7.4, 150 mM NaCl), and supplemented with PMSF and DNase I at a volume of 100 ml of buffer per 1 l of culture. All purification steps were carried out at 4°C. Cells were lysed using a high-pressure lyser (HPL6) at 1–1.5 kbar, and the lysate was centrifuged at 8,000 g for 30 min. Clarified lysate was supplemented with 30 mM Imidazole and incubated with 4 ml 50% Ni-NTA slurry per 1 l of culture for 2 hr under gentle agitation. Beads were washed with 25 CV TBS + 50 mM imidazole, and protein was eluted with 5 CV TBS containing 300 mM Imidazole. The elution was concentrated with a 10 kDa cut-off concentrator, filtered through a 0.22 μm filter, and loaded on a Superdex 75 10/300 GL column in TBS. Protein containing fractions were concentrated to 500 μM, frozen in liquid $N_2$, and stored at $-80°C$.

## Caspase-3 cleavage

One unit of recombinant human caspase-3 cleaves 1 nmol of its synthetic substrate at 37°C in 1 hr. In case of rXKR9, the cleavage was performed at 4°C overnight, using two units of caspase-3 per 1 nmol of purified rXKR9. To ensure caspase-3 activity, 10 mM DTT was supplemented. Caspase-3-cleaved protein (MW 41.6 kDa) was analyzed by ESI-MS by the Functional Genomics Center

Zurich (FGCZ) of the UZH to confirm successful cleavage. Briefly, protein was precipitated in a 30% aqueous solution of TCA. The pellet was washed two times in cold acetonitrile, before being dissolved in hexfluoroisopropanol and diluted 1:1 with a mixture of MeOH, 2PrOH, and 0.2% formic acid (FA) at a volume ratio of 30:20:50. This solution was further diluted with MeOH:2PrOH:0.2%FA (30:20:50), infused through a fused silica capillary, and sprayed through a PicoTip. The spray voltage was set to 3 kV, the cone voltage was set to 50 V, and the source temperature was set to 80°C. Nano-ESI-MS analysis was performed on a Synapt G2_Si mass spectrometer, and the data was recorded with MassLynx 4.2. Mass spectra were acquired in the positive-ion mode by scanning the m/z range from 400 to 5000 Da with a scan duration of 1 s and an interscan delay of 0.1 s. The recorded m/z data was deconvoluted into mass spectra by applying the maximum entropy algorithm MaxEnt1.

## SEC-MALS

SEC-MALS was performed to assess the oligomeric state of rXKR9. Ten micrograms of of purified rXKR9 was filtered through a 0.1 μm filter, injected onto a Superdex 200 Increase 3.2/300 GL column, equilibrated in LMNG SEC buffer, and run at room temperature on an Agilent 1260 Infinity II HPLC coupled with an Eclipse3 system equipped with a miniDAWN TREOS MALS detector and Optilab T-rEX refractometer (Wyatt Technology). Data was analyzed in the ASTRA software package. The *dn/dc* values used were 0.19 ml/g for rXKR9 and 0.132 ml/g for LMNG.

## Liposome reconstitution and scrambling assay

Chloroform-solubilized soybean polar extracts (79.5%) and tail-labeled NBD-PE or NBD-PS (0.5%, wt/wt) were pooled with cholesterol (20%, mol/mol). Chloroform was evaporated under a nitrogen stream, and lipids were washed once in chloroform and once in diethyl ether before drying in a desiccator overnight. The next day, the lipid film was solubilized in liposome buffer (20 mM HEPES pH 7.5, 300 mM KCl, 2 mM EGTA), supplemented with 35 mM CHAPS to a concentration of 20 mg/ml. Solubilized lipids were aliquoted, flash-frozen in liquid $N_2$, and stored at −80°C. Other investigated lipid compositions included 100% soybean polar lipids, 3 POPC:1 POPG (wt/wt), and 3 POPE:1 POPG (wt/wt). The lipids were prepared as described above.

On the day of purification, solubilized lipids were thawed at room temperature and diluted to 4 mg/ml in liposome buffer. Purified full-length or caspase-3-cleaved rXKR9 was added to lipids at a lipid to protein ratio of 100:1 (wt/wt, resulting in a molar ratio of 6000 lipid molecules per protein) and incubated at room temperature for 15 min under gentle agitation. For empty liposome controls, an equal amount of SEC buffer was added to the lipids instead of protein. For detergent removal and formation of proteoliposomes, SM-II bio-beads (20 mg per 1 mg lipids) were added to the lipid–protein mix. After 30 min of incubation under constant agitation at room temperature, another batch of bio-beads was added and the mix was transferred to 4°C. After 1 hr, the third fraction of bio-beads was added and the reconstitution mix was incubated at 4°C overnight while constantly agitating. The next morning, a final batch of bio-beads was added and incubated for 4 hr at 4°C. Proteoliposomes were transferred to room temperature and filtered through gravity-flow columns to remove the bio-beads. To harvest the proteoliposomes, samples were centrifuged at 170,000 g for 30 min at 22°C. Pelleted proteoliposomes were re-suspended in liposome buffer to 10 mg/ml and either flash-frozen in liquid $N_2$ and stored at −80°C or incubated with caspase-3 and 1 mM TCEP at room temperature overnight. After caspase-3 treatment, the proteoliposomes were flash-frozen and stored at −80°C. All steps were carried out in the dark to prevent bleaching of NBD fluorophores.

The scrambling measurements were essentially performed as described for TMEM16 scramblases without the addition of calcium (*Alvadia et al., 2019*; *Brunner et al., 2014*; *Kalienkova et al., 2019*). On the day of measurement, the proteoliposomes were thawed at room temperature, subjected to two freeze-thaw cycles and extruded 21 times through a 200 nm polycarbonate membrane, using an Avanti MiniExtruder. Extruded proteoliposomes were diluted in 80 mM HEPES pH 7.5, 300 mM KCl, 2 mM EGTA to 0.2 mg/ml, and initial fluorescence was recorded using a Spectrofluorometer (Ex.: 470 nm, Em.: 530 nm). After 60 s, 30 mM sodium dithionite was added and the measurement was continued for 360 s. The fluorescent decay was plotted as $F/F_{max}$. Reconstitution efficiency was examined by re-extraction of the protein with 1% DDM for 1 hr on ice. Extracted protein was injected onto a Superdex 200 Increase 5/150 GL column, equilibrated in SEC buffer.

## Characterization of mTMEM16F

mTMEM16F was expressed, purified, and reconstituted into liposomes as described in *Alvadia et al., 2019*. Briefly, mTMEM16F containing a C-terminal 3C cleavage site, myc and SBP tags, was expressed in stably transfected mTMEM16F HEK293T cells by tetracycline induction (2 μg/ml) for 48 hr. Harvested cells were re-suspended in lysis buffer (20 mM HEPES pH 7.5, 150 mM NaCl, 5 mM EGTA, 2% digitonin, and protease inhibitors), extracted for 2 hr, and centrifuged at 85,000 g for 30 min. The supernatant was incubated for 2 hr at 4°C with streptavidin UltraLink resin (1 ml beads per 20 ml cell pellet). The resin was washed with 60 CV SEC buffer (20 mM HEPES pH 7.5, 150 mM NaCl, 5 mM EGTA, 0.1% digitonin). Protein was eluted with 3 CV of SEC buffer, supplemented with 4 mM biotin, concentrated using 100 kDa cut-off concentrator, filtered through a 0.22 μm filter, and injected onto a Superose 6 10/300 GL column, pre-equilibrated in SEC buffer. Peak fractions were collected, concentrated, and used for liposome reconstitution. mTMEM16F was reconstituted into liposomes composed of 79.5% soybean polar lipid extracts, 20% (mol/mol) cholesterol, and 0.5% (wt/wt) 18:1-06:0 NBD-PE, at a 1:100 protein-to-lipid ratio (wt/wt). Harvested liposomes were re-suspended in liposome buffer, supplemented with the correct amount of $Ca(NO_3)_2$ to obtain the desired concentrations of free $Ca^{2+}$ as calculated with the online WEBMAXC calculator (*Bers et al., 2010*). After thawing and extrusion through a 400 nm membrane, the proteoliposomes were diluted in 80 mM HEPES pH 7.5, 300 mM KCl, 2 mM EGTA, plus $Ca^{2+}$, aiming for symmetric buffer conditions. NBD fluorescence was recorded as described for rXKR9.

## XKR8/9-basigin co-purification

To investigate the protein-protein interaction of rXKR9 and rat basigin, rXKR9 in pcDxNSM3 and rat basigin in pcDxC3GM were co-expressed and -purified. hXKR8 and human basigin in the same vectors were used as a positive control (*Suzuki et al., 2016*). For transfection of adherent HEK293T cells, the respective XKR and basigin DNA was mixed at a 1:2 ratio and then incubated with PEI MAX 40 kDa in a 1:2.5 ratio. The DNA sample diluted in non-supplemented DMEM was incubated at room temperature for 15 min and then added to HEK293T cells. Immediately after addition of the transfection mix, 3.5 mM valproic acid was added. Cells were incubated at 37°C and 5% $CO_2$. Forty-eight hours post-transfection, cells were harvested, washed in PBS, flash-frozen in liquid $N_2$, and pellets stored at −80°C.

XKR-basigin complexes were essentially purified as described in the protein purification section. HRV 3C cleavage was not performed to keep the fluorescent tag. In-gel fluorescence was monitored with a Viber Fusion FX7 imaging system (Ex.: 515 nm, Em.: 530 nm). Silver staining was performed according to manufacturer's instructions.

## Protein biotinylation

For sybody selection and ELISA, purified rXKR9 was chemically biotinylated using EZ-link NHS-PEG4-biotin. rXKR9 was concentrated to 2 mg/ml and incubated with a 10 times molar excess of EZ-link NHS-PEG4-biotin for 1 hr on ice. The amine-reactive reaction was quenched upon addition of 5 mM Tris–HCl pH 7.5. Excess biotin was separated from biotinylated protein on a PD-10 desalting column. Eluted protein was concentrated to 1.3 mg/ml, mixed with glycerol to a final concentration of 10%, aliquoted, flash-frozen in liquid $N_2$, and stored at −80°C.

## Sybody selection and ELISA

The sybody selection was performed as described previously (*Zimmermann et al., 2020*). The synthetic mRNA libraries, termed concave (CC), loop (LP), and convex (CX), and vectors required for selections were generously provided by Prof. Markus Seeger, Institute of Medical Microbiology, UZH. Selection and ELISA were performed in described buffers with the addition of LMNG where indicated. Briefly, one round of ribosome display was performed using the three different mRNA libraries, each encoding for $10^{12}$ different binders. The output library of ribosome display was cloned into a phagemid vector, which was used for two successive rounds of phage display. During the second round of phage display, an off-rate selection was performed to remove binders with a high off-rate. Eluted phages were quantified by qPCR, which allowed estimation of the enrichment. Enrichment was calculated based on the ratio of eluted phages from rXKR9 divided by the elution from the negative control (biotinylated TM287/288), which was purified as described previously

(*Hutter et al., 2019*). Enrichment factors after the second round of phage display were 1.7, 8.3, and 1.1 for CC, LP, and CX libraries. Despite the low enrichment factors, the libraries were subcloned by FX-cloning into pSb_init and expressed in 96-well plates. The periplasmic extract was subjected to ELISA in a 384-well format, using 50 µl of biotinylated rXKR9 or TM287/288 (50 nM/well). ELISA hits were analyzed by Sanger sequencing (Microsynth). Of 64 hits, 21 clones were further characterized for expression and biochemical behavior.

## Cryo-EM sample preparation and data collection

For the uncleaved cryo-EM sample, rXKR9 was purified as described above, except that purified sybody Sb1$^{XKR9}$ was added to rXKR9 prior to injecting onto the Superdex 200 10/300 GL column at 1.5 times molar excess. The SEC-purified complex was concentrated using a 50 kDa cut-off concentrator to 1.5 mg/ml and applied to glow-discharged (0.39 mbar, 15 mA, 30 s) holey carbon grids (Quanti-Foil R1.2/1.3 Au 200 mesh). Excess sample was removed by blotting for 2–4 s with zero blotting force in a controlled environment (4°C, 100% relative humidity) using a Vitrobot Mark IV. The sample was plunge-frozen in a liquid ethane-propane mix and stored in liquid $N_2$ for data collection.

For the cleaved cryo-EM sample, rXKR9 was purified as described above. After SEC, the protein was concentrated to 0.87 mg/ml and incubated with 24 units of caspase-3 and 10 mM DTT overnight on ice. The next-day, purified sybody Sb1$^{XKR9}$ was added at a 1.5 times molar excess, and complexes were subjected to SEC on a Superdex 200 10/300 GL column pre-equilibrated in SEC buffer. The cleaved complex was concentrated to 1.5 mg/ml and plunge-frozen the same way as the uncleaved sample.

The hXKR8-hBSG complex was purified as described above. The complex was concentrated to 2 mg/ml and plunge-frozen the same way as the rXKR9 samples. All rXKR9 samples were imaged on a Titan Krios G3i, operated at 300 kV, with a 100 µm objective aperture. All data were recorded using a post-column BioQuantum energy filter with a 20 eV slit and a K3 direct electron detector in super-resolution mode. Dose-fractionated micrographs were recorded with a defocus range of −1 µm to −2.4 µm in an automated mode using EPU 2.9. The datasets were acquired at a nominal magnification of 130,000×, corresponding to a pixel size of 0.651 Å per pixel (0.3255 Å per pixel in super-resolution mode), with 1.01 s exposure (36 frames) and a dose of 1.94 and 1.93 e⁻/Å²/frame for the datasets of uncleaved and cleaved protein, respectively. This resulted in a total electron dose of approximately 70 e⁻/Å² on the level of the specimen.

The hXKR8-BSG sample was imaged on a 300 kV Tecnai G² Polara microscope, with a 100 µm objective aperture. All data were collected using a post-column quantum energy filter with a 20 eV slit and a K2 Summit direct electron detector in counting mode. SerialEM 3.7 was used to set up automated data collection of dose-fractioned micrographs, with a defocus range of −0.8 µm to −2.5 µm. The dataset was recorded at a nominal magnification of 130'000 x, corresponding to a pixel size of 1.34 Å per pixel, with a total exposure of 12.4 s (31 frames) and a dose of 1.1 e⁻/Å²/frame, which resulted in a total electron dose of approximately 35 e⁻/Å² on the level of the specimen.

## Cryo-EM data processing

Data processing of XKR9 was performed in cryoSPARC v.3.0.1 and v.3.2.0 (*Punjani et al., 2017*). For the dataset of uncleaved protein, 12,396 micrographs were collected and subjected to patch motion correction and patch CTF estimation. Micrographs were Fourier-cropped once during motion correction, resulting in a pixel size of 0.651 Å per pixel. Low-quality micrographs were discarded based on the estimated resolution of CTF fits, relative ice thickness, and total full-frame motion, resulting in 9,823 good micrographs used for further analysis. A small subset of micrographs was used for particle picking, using a blob with 50 Å–120 Å diameter. Particles were extracted with a box size of 360 pixels and Fourier-cropped to 180 pixels (1.302 Å per pixel). From two rounds of 2D classification with marginalization over poses and shifts and a circular 150 Å mask, 2D class averages showing protein features were used as templates for automated template-based picking from all high-quality micrographs. Particles were extracted as described above. After two rounds of 2D classification, 2,171,713 particles were subjected to an ab initio reconstruction with five classes, no similarity, C1 symmetry, and initial alignment resolution of 12 Å. 1,852,344 particles, from classes with protein features, were directed into a second ab initio reconstruction with three classes, no similarity, C1 symmetry, and 7 Å initial resolution. The best-resolved class, together with the worst-resolved class from

the first ab initio reconstruction, were used as references for three rounds of heterogeneous refinement, together with the particle set selected from the first round of ab initio reconstruction. After heterogeneous refinement, where the initial resolution of the input volumes was set to 7 Å, 866,439 particles of high quality were initially used in a non-uniform (NU) refinement (*Punjani et al., 2020*) and afterwards locally CTF refined. Subsequently, the particle alignment was further improved in a second NU-refinement step using C1 symmetry, yielding a map with a resolution of 3.66 Å according to the 0.143 cut-off criterion (*Rosenthal and Henderson, 2003*). During NU-refinement, the resolution of the input structure was filtered to 12 Å and per-group CTF parameters were optimized. The directional resolution was estimated with the 3DFSC server (*Tan et al., 2017*), and local resolution estimation was performed within cryoSPARC.

For cleaved rXkr9, 14,929 micrographs were collected and motion-corrected, CTF-estimated and curated as described for the micrographs from the dataset of the uncleaved sample. This procedure resulted in 12,135 of high-quality micrographs, that were used for template-based picking using templates created from the 3.66 Å map from the dataset of uncleaved protein. Particles were extracted as described previously. After two rounds of 2D classification, 834,105 and 496,296 particles were used for a first and second ab initio reconstruction with the same parameters as for the dataset of uncleaved protein. All particles used for the second round of ab initio reconstruction were subjected to five rounds of heterogeneous refinement, using the same parameters as described above. The best-resolved class from the second ab initio reconstruction and a decoy class, generated from deselected particles from 2D classifications, were used as input volumes. Analogously, particle alignment was improved by NU-refinement, local CTF refinement, and a second step of NU-refinement yielding a map with a resolution of 4.3 Å.

The hXkr8-hBSG data was processed in RELION 3.0.7 (*Zivanov et al., 2018*). Two thousand two hundred and twelve micrographs were collected and motion-corrected using MotionCor2 v.1.2.3 (*Zheng et al., 2017*). After CTF estimation performed with CTFFIND4.1 (*Rohou and Grigorieff, 2015*), micrographs with a resolution lower than 6 Å were discarded, resulting in 2,054 of high-quality micrographs. 1,000 particles were manually picked, extracted with a box size of 160 pixels, binned to 80 pixels (2.68 Å per pixel), and subjected to an initial 2D classification. 2D classes showing protein features were used for auto-picking. 465,373 particles were extracted as described above and directed into four rounds of 2D classification. 282,197 particles were used to generate an initial 3D model and then subjected to one round of 3D classification. The most promising class was refined in a 3D refinement, yielding a low-resolution structure at 15.3 Å.

## Model building and refinement

The models of full-length rXKR9 and caspase-cleaved rXKR9 were built in Coot (*Emsley and Cowtan, 2004*). Full-length rXKR9 was built de novo into the cryo-EM density, and cleaved rXKR9 was built using the full-length structure as reference. The cryo-EM density of full-length rXKR9 was of sufficiently high resolution to unambiguously assign residues 1–66, 81–105, 119–344, and 365–373. Sb1$^{XKR9}$ was initially modeled based on the high-resolution structure of a nanobody (PDB: 1ZVH), with CDR3, which interacts with the membrane protein, being rebuilt into the cryo-EM density. The atomic models were improved iteratively by cycles of real-space refinement in PHENIX (*Afonine et al., 2018*), with secondary structure constraints applied followed by manual corrections in Coot. Validation of the models was performed using PHENIX real-space refinement against half-map A after applying random shifts of 0.3 Å to the final model (FSC$_{work}$). FSC$_{free}$ was calculated between the shifted model and half-map B and the overall map-to-model FSC between the final model and the full map (FSC$_{sum}$). Surfaces were calculated with MSMS (*Sanner et al., 1996*). Figures and videos containing molecular structures and densities were prepared with DINO (http://www.dino3d.org), Chimera (*Pettersen et al., 2004*), and ChimeraX (*Pettersen et al., 2021*).

## Acknowledgements

This work was supported by a grant of the European Research Council (ERC no 339116, AnoBest) to Raimund Dutzler and a Candoc grant of the Forschungskredit of the University of Zurich (grant no FK-20–040) to MSS. The contribution of Janine D Brunner and Stephan Schenck to the preparation and evaluation of XKR constructs prior to the described study is acknowledged. The cryo-electron microscope and K3-camera were acquired with support of the Baugarten and Schwyzer-Winiker

foundations and a Requip grant of the Swiss National Science Foundation. We thank Simona Sorrentino and the Center for Microscopy and Image Analysis (ZMB) of the University of Zurich for the support and access to the electron microscope and S Rutz for help during data collection and Markus A Seeger for providing the sybody libraries. Mass spectrometry analysis was carried out with the help of the Functional Genomics Center Zurich (FGCZ). All members of the Dutzler lab are acknowledged for their help at various stages of the project.

## Additional information

### Funding

| Funder | Grant reference number | Author |
|---|---|---|
| FP7 European Research Council | 339116, AnoBest | Raimund Dutzler |
| University of Zurich | FK-20-040 | Monique S Straub |

The funders had no role in study design, data collection and interpretation, or the decision to submit the work for publication.

### Author contributions

Monique S Straub, Conceptualization, Data curation, Formal analysis, Validation, Investigation, Visualization, Methodology, Writing - original draft, Writing - review and editing; Carolina Alvadia, Investigation, Methodology, Writing - review and editing; Marta Sawicka, Validation, Methodology, Writing - review and editing; Raimund Dutzler, Conceptualization, Supervision, Funding acquisition, Validation, Investigation, Visualization, Writing - original draft, Project administration, Writing - review and editing

### Author ORCIDs

Monique S Straub ⓘ https://orcid.org/0000-0002-7721-5048
Carolina Alvadia ⓘ http://orcid.org/0000-0001-8446-1098
Marta Sawicka ⓘ http://orcid.org/0000-0003-4589-4290
Raimund Dutzler ⓘ https://orcid.org/0000-0002-2193-6129

### Decision letter and Author response

Decision letter https://doi.org/10.7554/eLife.69800.sa1
Author response https://doi.org/10.7554/eLife.69800.sa2

## Additional files

### Supplementary files

• Source data 1. Cumulated source data.

• Transparent reporting form

### Data availability

Coordinates have been deposited in the PDB under accession codes 7P14 and 7P16. Cryo-EM maps have been deposited in the EMBD under accession codes EMDB-13155 and EMDB-13157. Source data files have been provided for Figure 1A-E,G, Figure 1-figure supplements 1 and 3B-D, Figure 2-figure supplements 1E and 2E.

The following datasets were generated:

| Author(s) | Year | Dataset title | Dataset URL | Database and Identifier |
|---|---|---|---|---|
| Straub MS, Sawicka M, Dutzler R | 2021 | Structure of full-length rXKR9 in complex with a sybody at 3.66A | https://www.rcsb.org/structure/7P14 | RCSB Protein Data Bank, 7P14 |

| Straub MS, Sawicka M, Dutzler R | 2021 | Cryo-EM density of full-length rXKR9 in complex with a sybody at 3.66A | https://www.ebi.ac.uk/pdbe/entry/emdb/EMD-13155 | Electron Microscopy Data Bank, EMD-13155 |
|---|---|---|---|---|
| Straub MS, Sawicka M, Dutzler R | 2021 | Structure of caspase-3 cleaved rXKR9 in complex with a sybody at 4.3A | https://www.rcsb.org/structure/7P16 | RCSB Protein Data Bank, 7P16 |
| Straub MS, Sawicka M, Dutzler R | 2021 | Cryo-EM density of caspase-3 cleaved rXKR9 in complex with a sybody at 4.3A | https://www.ebi.ac.uk/pdbe/entry/emdb/EMD-13157 | Electron Microscopy Data Bank, EMD-13157 |

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
