## [Decision Letter]

**Acceptance summary:**

This paper reports the atomic structure of XKR9, a membrane protein that is implicated in initiating the process to get rid of cells that are undergoing programmed cell death (apoptosis). The protein of interest was originally proposed to be a lipid channel, but the work presented here suggests that it is unlikely to function in this capacity alone. Still, the structural work presented is significant, and altogether this study represents an important step forward in this nascent field.

**Decision letter after peer review:**

Thank you for submitting your article "Cryo-EM structures of the Caspase activated protein XKR9 involved in apoptotic lipid scrambling" for consideration by *eLife*. Your article has been reviewed by 3 peer reviewers, one of whom is a member of our Board of Reviewing Editors and the evaluation has been overseen by Olga Boudker as the Senior Editor. The following individual involved in review of your submission have agreed to reveal their identity: Anant K Menon (Reviewer #3).

Essential Revisions:

There was general enthusiasm for this paper, with the work regarded as being of high quality, and a significant structural advance in a nascent field. We have the following requests for revisions.

1) Please remove the summary Figure 7 and limit the speculative narrative about molecular mechanism of XKR9 in the Results and Discussion sections. Subsection titles should be modified to provide an objective interpretation of the results. For example, "Features of potential functional relevance" could be changed to something like "Analysis of putative lipid pathways". Similarly "Activation" can be changed to something like "Caspase cleavage of the C-terminus", which provides a more direct presentation of the experiment conducted. With this, the paper can focus on the novelty of the structures obtained, discussing the new architecture and evidence of lipids in the protein in XKR9.

2) All three reviewers made the following comment: In Figure 1, no activity can be detected from full-length or cleaved XKR9 reconstituted into proteoliposomes. As controls, scramblase experiments were performed with proteoliposomes containing TMEM16F in the presence or absence of ca^2+^. While this is a reasonable control, a low-resolution structure of XKR8 in complex with its accessory protein basigin is presented in the same figure demonstrating that it is biochemically stable. Thus, it would be helpful to perform the scramblase assay with XKR8/basigin, which is hypothesized to be a fully functioning member of the same family. This would provide evidence that the assay, which is well-established for TMEM16 family members, is also suitable for detecting activity by XKR scramblases in vitro. While the observation of function in XKR8/basigin would represent a significant advance, it is not required for publication. Still, carrying out this experiment, and discussion of these efforts would help to smooth the somewhat awkward transition between the presentation of these two proteins in the current manuscript.

3) Please respond to the comments in the recommendations to authors sections below.

*Reviewer #1 (Recommendations for the authors):*

1. The lack of functional activity is a key question here, and it is understandable that the reconstituted system may not contain the required components. However, since PS is the main lipid that gets scrambled in cells, have you examined higher PS lipid compositions other than the small amount of labelled PS that you have currently? Alternatively, have you tested the function of membrane vesicles to confirm that the protein is functional when expressed?

2. Page 7, line 138: It is concluded that basigin is not the required component for scrambling because it does not form a complex with XKR9 in your experiments, and thus does not confer scrambling. However, it seems possible that it could form a complex under different conditions. It is also possible that XKR9 is the component that is not in a proper state to form a complex with basigin, especially considering that XKR9 is not functions. Many different options still seem possible and so this conclusion is not well supported.

3. In the SDS-PAGE images, the shift in the caspase-3 treated bands is not clear. Assuming that you are trying to show that caspase-3 successfully cleaved this fragment, these results should be optimized on a different acrylamide gel to show that the shift is clearly discernable.

4. Figure 1D – "Data show mean and standard deviations of three technical replicates." This panel appears to contain the raw traces of the fluorescent quenching, not the mean {plus minus} st.dev. of the traces. Is this correct?

5. Please report the molecular weights of the different protein species in the methods.

6. Please report the lipid reconstitution conditions in mole fraction units (i.e. 1 monomer per 10000 lipids) in addition to the 1:100 wt/wt ratio.

7. Page 6, line 117: Please clarify about the inside-out liposomes, was this just by random orientation or were inside-out liposomes purified directly?

8. There were several typos found in the document. Please read through and correct others:

Typo on page 6, line 95: "multi angle" should be "multi-angle"

Typo on page 17, line 346: "… its structure structures do not provide…"

Typo on page 24, line 473: "TMEM16scramblases"

9. As a reminder, the PDBs will need to be deposited and listed in the Accession code section of the paper.

*Reviewer #2 (Recommendations for the authors):*

1. Density attributed to a phospholipid is present in both structures. Despite its potential importance to the protein's functional role, there is little mention of the interactions between the lipid and XKR9 in the text and no detailed figure showing the interactions. I would suggest that additional efforts be made in describing the lipid-binding site and its potential role in scramblase activity.

2. Continuing from point 2, were non-protein densities resolved in the C2 or C3 cavities of either structure?

---

## [Author Response]

Essential Revisions (for the authors):There was general enthusiasm for this paper, with the work regarded as being of high quality, and a significant structural advance in a nascent field. We have the following requests for revisions.1) Please remove the summary Figure 7 and limit the speculative narrative about molecular mechanism of XKR9 in the Results and Discussion sections. Subsection titles should be modified to provide an objective interpretation of the results. For example, "Features of potential functional relevance" could be changed to something like "Analysis of putative lipid pathways". Similarly "Activation" can be changed to something like "Caspase cleavage of the C-terminus", which provides a more direct presentation of the experiment conducted. With this, the paper can focus on the novelty of the structures obtained, discussing the new architecture and evidence of lipids in the protein in XKR9.

In our revised manuscript, we have removed Figure 7 and we have introduced the suggested changes in the titles of the subsections. We have also shortened the speculations about potential molecular mechanisms in the discussion.

2) All three reviewers made the following comment: In Figure 1, no activity can be detected from full-length or cleaved XKR9 reconstituted into proteoliposomes. As controls, scramblase experiments were performed with proteoliposomes containing TMEM16F in the presence or absence of Ca^2+^. While this is a reasonable control, a low-resolution structure of XKR8 in complex with its accessory protein basigin is presented in the same figure demonstrating that it is biochemically stable. Thus, it would be helpful to perform the scramblase assay with XKR8/basigin, which is hypothesized to be a fully functioning member of the same family. This would provide evidence that the assay, which is well-established for TMEM16 family members, is also suitable for detecting activity by XKR scramblases in vitro. While the observation of function in XKR8/basigin would represent a significant advance, it is not required for publication. Still, carrying out this experiment, and discussion of these efforts would help to smooth the somewhat awkward transition between the presentation of these two proteins in the current manuscript.

In our study, we have employed the described in vitro assay to monitor lipid transport by XKR9. We and others have previously used the same assay to characterize different lipid scramblases and we thus think that it is generally suitable to detect protein-catalyzed lipid movement between both leaflets of the bilayer. In our experiments, we have included data of TMEM16F as a proper control to illustrate the range of the expected change of the signal between an inactive and an active scramblase. This is particularly important since the basal fluorescence level does to some degree vary upon protein reconstitution, even in case a protein does not itself mediate lipid flip-flop.

As described in our manuscript, our data has not provided strong evidence for the reconstituted XKR9 to act as a caspase-activated lipid scramblase. Besides XKR9, we have also carried out equivalent experiments for an XKR8-basigin complex and found very similar results (Author response image 1). Although the data from cellular assays, which suggest that XKR proteins themselves form the scrambling units, are strong, this activity could so far not be detected for purified and reconstituted protein in vitro. This leaves the question whether isolated XKR proteins would be sufficient to catalyze lipid movements unanswered. At this stage, we cannot exclude the possibility that the assay in its current form is not suitable to characterize scrambling activity of purified and reconstituted XKR proteins. Since, in contrast to XKR9, the experiments on the XKR8-basigin complex were only performed once and XKR8 is not the main subject of our investigation, we refrain from including the data in the manuscript.

**Author response image 1. respfig1:** Lipid scrambling assay of the XKR8-basigin complex. (A) Size exclusion chromatogram of proteins extracted from proteoliposomes containing a full-length reconstituted XKR8-basigin complex (uncleaved), the same proteoliposomes treated with caspase-3 added to the surrounding media (assuming that a fraction of the cleavage sites would be accessible from the outside, liposome cleaved) or protein that was incubated with caspase 3 in solution overnight prior to reconstitution (buffer cleaved). The peaks corresponding to the non-aggregated protein complex is indicated (#). (B) Assay to monitor the protein-catalyzed movement of fluorescent lipids between both leaflets of a bilayer. The addition of the reducing agent dithionite to the solution (*) is indicated. Traces of TMEM16F in absence and presence of 100 µM ca^2+^ are shown as controls for an inactive and active scramblase, respectively. No pronounced activity is observed in any of the samples with either full-length or caspase-3-treated XKR8-basigin proteoliposomes. (C) Traces of XKR8-basigin proteoliposomes shown in panel B in comparison to scrambling data of XKR9 shown in Figure 1D of the manuscript (dashed lines).

We have also rewritten the chapter describing the investigation of a potential interaction of XKR9 with the type 1 membrane protein basigin to remove any claim that it might be a necessary cofactor for scrambling. The aim of the experiments related to the co-expression of XKR8 and XKR9 with basigin was to show whether we would be able to detect a basigin interaction with XKR9, which was not the case. In these experiments, XKR8 was used as control to show the validity of the approach. Although the interaction of XKR8 with basigin is now established (see also the recent structure of a XKR8-basign complex by the Nagata and Toyoshima groups deposited in bioRxiv (Sakuragi et al., 2021)), the impact of basigin on XKR8 function remains unclear.

3) Please respond to the comments in the recommendations to authors sections below.Reviewer #1 (Recommendations for the authors):1. The lack of functional activity is a key question here, and it is understandable that the reconstituted system may not contain the required components. However, since PS is the main lipid that gets scrambled in cells, have you examined higher PS lipid compositions other than the small amount of labelled PS that you have currently? Alternatively, have you tested the function of membrane vesicles to confirm that the protein is functional when expressed?

We have not tested the function of the protein in membrane vesicles. In light of the basal expression of apoptotic scramblases such assays do either require the use of genetic knock-outs or a near quantitative downregulation by siRNA. With respect to the preparation of liposomes, we have experimented with different lipid compositions and found the used soybean-cholesterol mixture to be compatible with proper reconstitution. It should be emphasized that this source would provide a lipid composition that is close to the content of eukaryotic cells. The same lipid composition was used for the reconstitution and characterization of the scramblase TMEM16F, which is fully activatable and shown as control (Figure 1D, Figure 1—figure supplement 1C). Although not explicitly listed in the manufacturer’s specification, PS is a lipid that is found in soybeans and we thus expect it to be present in our samples beyond the 0.5% of labeled lipids added to our reconstitution. We also want to emphasize that, although the biological response is evoked by the exposure of PS to the outside, scramblases are generally not considered as specific and their transport is usually bidirectional. Although this poor selectivity has not yet been explicitly demonstrated for apoptotic scramblases, it is well established for scramblases of the TMEM16 family.

2. Page 7, line 138: It is concluded that basigin is not the required component for scrambling because it does not form a complex with XKR9 in your experiments, and thus does not confer scrambling. However, it seems possible that it could form a complex under different conditions. It is also possible that XKR9 is the component that is not in a proper state to form a complex with basigin, especially considering that XKR9 is not functions. Many different options still seem possible and so this conclusion is not well supported.

Our motivation of the experiment was to investigate whether basigin would form a similar robust complex with XKR9 as previously demonstrated for XKR8. In our experiments, XKR8 served as a positive control for which we were able to detect complex formation in a straightforward fashion. In contrast, no sign of interaction was found under equivalent conditions for XKR9. We did not want to imply that basigin would itself play a direct role in lipid scrambling in XKR8, which we do not know at this stage. We have clarified this in our revised manuscript.

3. In the SDS-PAGE images, the shift in the caspase-3 treated bands is not clear. Assuming that you are trying to show that caspase-3 successfully cleaved this fragment, these results should be optimized on a different acrylamide gel to show that the shift is clearly discernable.

We would like to emphasize that we have frequently quantified the extent of caspase cleavage by mass spectrometry as shown for the sample of the XKR9-Sb1^XKR9^ complex used for structure determination (Figure 1—figure supplement 3D). The displayed gels generally correspond to the samples used for the experiments. The comparison between full-length and cleaved protein in Figure 1B was not essential and we have removed the full-length band to only show the gel fraction used for the SEC-experiment. The gel shown Figure 1C corresponds to protein that was extracted from proteoliposomes used for the scrambling experiments shown in Figure 1D. In this case, the shift between uncleaved and cleaved protein is obvious as also illustrated in the plotted density profile shown (Author response image 2).

**Author response image 2. respfig2:** SDS PAGE gel showing mass differences between full-length and caspase-cleaved samples. Left, SDS PAGE gel of full-length and caspase-3 treated samples with the integrated lanes numbered and indicated by yellow boxes. Right, integrated density of the gel along the direction of migration. These bands are also displayed in Figure 1C of the manuscript. The difference in the migration of uncleaved samples (red) and cleaved samples (blue, green) is apparent.

4. Figure 1D – "Data show mean and standard deviations of three technical replicates." … This panel appears to contain the raw traces of the fluorescent quenching, not the mean {plus minus} st.dev. of the traces. Is this correct?

The panel shows the average of three technical replicates. The standard deviation shown in grey is so small that it barely extends beyond the linewidth of the curve.

5. Please report the molecular weights of the different protein species in the methods.

We have added the requested information in the methods.

6. Please report the lipid reconstitution conditions in mole fraction units (i.e. 1 monomer per 10000 lipids) in addition to the 1:100 wt/wt ratio.

We have added the requested information in the methods (1 monomer per 6’000 lipid molecules).

7. Page 6, line 117: Please clarify about the inside-out liposomes, was this just by random orientation or were inside-out liposomes purified directly?

Inside-out liposomes were not purified directly. The reconstitution was performed with a protocol using solubilized lipids, which should in principle allow a random orientation of the protein. The observed preference for an inside-out orientation was a consequence of the sample properties.

8. There were several typos found in the document. Please read through and correct others:Typo on page 6, line 95: "multi angle" should be "multi-angle"Typo on page 17, line 346: "… its structure structures do not provide…"Typo on page 24, line 473: "TMEM16scramblases"

We have corrected these mistakes.

9. As a reminder, the PDBs will need to be deposited and listed in the Accession code section of the paper.

We have deposited the coordinates with the PDB and listed the accession codes in the revised manuscript.

Reviewer #2 (Recommendations for the authors):1. Density attributed to a phospholipid is present in both structures. Despite its potential importance to the protein's functional role, there is little mention of the interactions between the lipid and XKR9 in the text and no detailed figure showing the interactions. I would suggest that additional efforts be made in describing the lipid-binding site and its potential role in scramblase activity.

We have extended the description of the bound lipid in site C1 in the revised manuscript and added Figure 5—figure supplement 1 to better describe its interaction with the protein.

2. Continuing from point 2, were non-protein densities resolved in the C2 or C3 cavities of either structure?

We also found residual density of two potential phospholipids in the cavity C2 and have documented this in Figure 5—figure supplement 1C and D. In both cases we have included models in the refined structure of the full-length protein.